# Contrastive Predictive Coding Done Right for Mutual Information Estimation

**J. Jon Ryu[1], Pavan Yeddanapudi[1], Xiangxiang Xu[2], Gregory W. Wornell[1]**
[1]Department of EECS, MIT, Cambridge, MA 02139, USA
[2]Department of Computer Science, University of Rochester, Rochester, NY 14627, USA
`{jongha,pky,gww}@mit.edu, xiangxiangxu@rochester.edu`

## Abstract

The InfoNCE objective, originally introduced for contrastive representation learning, has become a popular choice for mutual information (MI) estimation, despite its indirect connection to MI. In this paper, we demonstrate why InfoNCE should not be regarded as a valid MI estimator, and we introduce a simple modification, which we refer to as *InfoNCE-anchor*, for accurate MI estimation. Our modification introduces an auxiliary *anchor* class, enabling consistent density ratio estimation and yielding a plug-in MI estimator with significantly reduced bias. Beyond this, we generalize our framework using proper scoring rules, which recover InfoNCE-anchor as a special case when the log score is employed. This formulation unifies a broad spectrum of contrastive objectives, including NCE, InfoNCE, and $f$-divergence variants, under a single principled framework. Empirically, we find that InfoNCE-anchor with the log score achieves the most accurate MI estimates; however, in self-supervised representation learning experiments, we find that the anchor does not improve the downstream task performance. These findings corroborate that contrastive representation learning benefits not from accurate MI estimation per se, but from the learning of structured density ratios.

## 1 Introduction

Contrastive learning has become a cornerstone of modern unsupervised representation learning, powering advances in computer vision (Chen et al., 2020), natural language processing (Mikolov et al., 2013; Levy & Goldberg, 2014), and beyond (Jaiswal et al., 2020; Hu et al., 2024). One of the key ingredients in many contrastive methods is the *InfoNCE* objective (van den Oord et al., 2018). While originally proposed as a representation learning framework, van den Oord et al. (2018) noted that the InfoNCE objective can be used to evaluate mutual information (MI), interpreting the objective as a variational bound on MI; see (van den Oord et al., 2018, Appendix A). This interpretation has led to its widespread use for MI estimation, e.g., (Poole et al., 2019; Song & Ermon, 2020a; Gowri et al., 2024; Lee & Rhee, 2024). It is also widely known, however, that InfoNCE often yields a rather loose bound on MI (Poole et al., 2019; Tschannen et al., 2020). As a result, the InfoNCE estimator is generally considered a low-variance but high-bias MI estimator (van den Oord et al., 2018). Although several proposals have been made to address this issue since its inception, the effectiveness (i.e., the low-variance property) and the limitation (i.e., the high bias) of InfoNCE remain poorly understood.

In this paper, we clarify the operational meaning of the InfoNCE objective, by showing that the objective should be understood as a variational lower bound of a statistical divergence different from the mutual information. Building on this clarification, we establish a sharp characterization of its relationship to the Kullback–Leibler (KL) divergence, revealing why the InfoNCE objective should not be regarded as a direct estimate of MI. We further argue that InfoNCE can be viewed as a density ratio estimation objective, while the critic (or its exponentiated form) estimates the density ratio $\frac{p(x,y)}{p(x)p(y)}$ only up to an arbitrary function $C(y)$, rendering it unsuitable for use in a plug-in estimator.

To address this limitation, we introduce a simple modification to the variational objective, which we call *InfoNCE-anchor*, corresponding to an alternative divergence. In the new framework, the inclusion of an *anchor* enables the (exponentiated) critic to estimate the density ratio $\frac{p(x,y)}{p(x)p(y)}$ directly. This adjustment facilitates consistent density ratio estimation and yields a plug-in MI estimator that

retains the low variance of InfoNCE while significantly reducing its bias. See Figure 1 for a quick comparison with existing estimators, where the new plug-in estimator based on InfoNCE-anchor demonstrates low-bias, low-variance performance.

We generalize our framework using tools from statistical decision theory, showing that InfoNCE-anchor corresponds to the *log score*, a canonical example of a proper scoring rule. This insight reveals that many contrastive objectives, including NCE, InfoNCE, and certain $f$-divergence variants, can be unified under a single principled framework of density ratio estimation using proper scoring rules.

Empirically, we show that estimators induced by the log score yields state-of-the-art MI estimates across a range of settings. In contrastive representation learning tasks, however, we find that multiple scoring rules yield similar performance, suggesting that MI estimation is not the primary driver of contrastive learning success. Instead, our results support that contrastive learning benefits from learning structured density ratios, regardless of whether the objectives are accurate MI estimators.

While InfoNCE has played a canonical role in contrastive representation learning, its rather loose association to MI estimation has historically fostered the misconception that representation learning is essentially about maximizing MI; see, e.g., (Bachman et al., 2019; Wu et al., 2020). This paper clarifies why such an interpretation can be limiting and imprecise, and that contrastive representation learning should instead be framed as representation learned to factorize pointwise MI (PMI) $\log \frac{p(x,y)}{p(x)p(y)}$, or pointwise dependence (PD) $\frac{p(x,y)}{p(x)p(y)}$. All proofs can be found in Appendix G. All logarithms are in base 2; hence KL divergence and MI are measured in bits.

## 2 PRELIMINARIES

In this paper, we first review different types of variational-bounds-based MI estimators in the literature, and provide a taxonomy. We then delve into the InfoNCE estimator, and show why the InfoNCE estimator should not be considered as a direct estimate for MI.

### 2.1 TYPES OF INFORMATION ESTIMATORS

Existing variational-bound-based MI estimators can be categorized into three principal categories as follows, based on the relationship of their optimization objectives to the final metrics to compute MI. Table 1 summarizes the representative estimators.

- **Type 1: Training and evaluation with a single variational lower bound.** These estimators optimize a tractable lower bound on the MI and use the same bound for evaluation. Examples include DV (Donsker & Varadhan, 1975), NWJ (Nguyen et al., 2010), and InfoNCE (van den Oord et al., 2018). While conceptually simple and natural, McAllester & Stratos (2020) showed that any distribution-free high-probability lower bound of MI is upper bounded by $\log N$, where $N$ is the sample size. This result implies that variational lower-bound–based sample estimates of MI suffer from an inherent limitation.

- **Type 2: Training with a variational lower bound, evaluation by plugging-in to another variational lower bound.** These estimators optimize a surrogate objective, often smoothed or stabilized for optimization, and then estimate MI via plug-in to a different bound such as DV or NWJ. Examples include MINE (Belghazi et al., 2018), JS (Hjelm et al., 2019), and SMILE (Song & Ermon, 2020a). These methods often improve stability during training, but introduce additional sources of mismatch between optimization and evaluation. Note that the critique of McAllester & Stratos (2020) still applies to this type of estimators.

- **Type 3: Training with a variational lower bound, evaluation with a plug-in estimator.** These estimators target to learn the density ratio $\frac{p(x,y)}{p(x)p(y)}$ directly and compute MI by plugging the estimated score function into the definition of MI. This includes recent methods like PCC/D-RFC (Tsai et al., 2020), $f$-DIME (Letizia et al., 2024), and JSD-LB (Dorent et al., 2025), as well as our method to be proposed below. These approaches provide greater flexibility and potentially lower bias, side-stepping from the issue of the variational lower-bound approach, as they decouple density ratio learning from specific bounds.

Table 1: Overview of existing variational-bound-based MI estimators. In this table, we use the standard critic parametrization, which aims to train $c(x,y) \approx \log \frac{p(x,y)}{p(x)p(y)}$.

| | Estimator | Optimization objective $\mathcal{L}(c)$ (loss) | Estimator $\hat{I}(X;Y)$ |
|---|---|---|---|
| **Type 1** | DV (Donsker & Varadhan, 1975) | $\mathcal{L}_{\mathrm{DV}}(c) \triangleq -\mathbb{E}_{p(x,y)}[c(x,y)] + \log \mathbb{E}_{p(x)p(y)}[e^{c(x,y)}]$ | $-\mathcal{L}_{\mathrm{DV}}(c)$ |
| | NWJ (Nguyen et al., 2010) | $\mathcal{L}_{\mathrm{NWJ}}(c) \triangleq -\mathbb{E}_{p(x,y)}[c(x,y)] + \mathbb{E}_{p(x)p(y)}[e^{c(x,y)}] - 1$ | $-\mathcal{L}_{\mathrm{NWJ}}(c)$ |
| | InfoNCE (van den Oord et al., 2018) or NT-XEnt (Chen et al., 2020) | $\mathcal{L}_{\mathrm{InfoNCE}}(c) \triangleq -\mathbb{E}_{p^K(x,y)}\left[ \frac{1}{K} \sum_{i=1}^{K} \log \frac{c(x_i,y_i)}{\frac{1}{K}\sum_{j=1}^{K} c(x_i,y_j)} \right]$ | $-\mathcal{L}_{\mathrm{InfoNCE}}(c)$ |
| | MLInfoNCE (Song & Ermon, 2020b) | $\mathcal{L}_{\mathrm{MLInfoNCE}}(\theta)$ (see Eq. (7)) | $-\mathcal{L}_{\mathrm{MLInfoNCE}}(\theta)$ |
| **Type 2** | MINE (Belghazi et al., 2018) | $\mathcal{L}_{\mathrm{MINE}}(c) \triangleq -\mathbb{E}_{p(x,y)}[c(x,y)] + \frac{\mathbb{E}_{p(x)p(y)}[e^{c(x,y)}]}{\mathrm{EMA}(\mathbb{E}_{p(x)p(y)}[e^{c(x,y)}])}$ | $-\mathcal{L}_{\mathrm{DV}}(c)$ |
| | JS (Poole et al., 2019) or NT-Logistics (Chen et al., 2020) | $\mathcal{L}_{\mathrm{JS}}(c) \triangleq \mathbb{E}_{p(x,y)}[\mathrm{sp}(-c(x,y))] + \mathbb{E}_{p(x)p(y)}[\mathrm{sp}(c(x,y))]$ | $-\mathcal{L}_{\mathrm{NWJ}}(c)$ |
| | SMILE (Song & Ermon, 2020a) | $\mathcal{L}_{\mathrm{JS}}(c)$ | $-\mathcal{L}_{\mathrm{clippedDV}}(c)$ |
| **Type 3** | PCC / D-RFC (Tsai et al., 2020) | $\mathcal{L}_{\mathrm{JS}}(c)$ / $\mathcal{L}_{\chi^2}(c) \triangleq -2\mathbb{E}_{p(x,y)}[e^{c(x,y)}] + \mathbb{E}_{p(x)p(y)}[e^{2c(x,y)}]$ | $\mathbb{E}_{\hat{p}(x,y)}[c(x,y)]$ |
| | $f$-DIME (Letizia et al., 2024) | $\mathcal{L}_{f\text{-NWJ}}(c)$ | $\mathbb{E}_{\hat{p}(x,y)}[c(x,y)]$ |
| | JSD-LB (Dorent et al., 2025) | $\mathcal{L}_{\mathrm{JS}}(c)$ | $\mathbb{E}_{\hat{p}(x,y)}[c(x,y)]$ |
| | **InfoNCE-anchor** (this paper) | $\mathcal{L}_{K;\nu}^{\Psi}(c)$ (see Eq. (4) and Eq. (13)) | $\mathbb{E}_{\hat{p}(x,y)}[c(x,y)]$ |

## 2.2 DEMYSTIFYING THE INFONCE ESTIMATOR

Despite its inception as an objective for contrastive representation learning (van den Oord et al., 2018), InfoNCE has become widely considered as a MI estimator. In this section, we revisit the analytical foundation of the objective and disentangle what InfoNCE is *claimed* to measure from what InfoNCE indeed characterizes. Our goal is two-fold: (1) reveal the divergence that InfoNCE targets, and (ii) quantify the precise gap between that divergence and the mutual information. Before we proceed, we remark that the core of InfoNCE can be better described when we contrast two abstract distributions $q_1(x)$ and $q_0(x)$, which can be replaced by $p(x|y)$ and $p(x)$, respectively, if we wish to specialize it for mutual information.

Throughout, let $x_1$ denote a *positive* example drawn from the data distribution $q_1$, and let $x_2, \ldots, x_K$ be *negative* examples drawn i.i.d. from a noise distribution $q_0$. We let $x_{i:j} \triangleq (x_i, \ldots, x_j)$ for $i \leq j$ as a shorthand. A score network (or critic) $r_\theta \colon \mathcal{X} \to \mathbb{R}_{>0}$ is trained to assign large values to real samples and small values to negatives, and the InfoNCE loss compares $r_\theta(x_1)$ against the arithmetic mean of $r_\theta(x_z)$ over the whole batch.

$$\mathcal{L}_{\mathrm{InfoNCE}}(\theta) \triangleq -\mathcal{D}_{\mathrm{InfoNCE}}(\theta) \triangleq -\mathbb{E}_{q_1(x_1)q_0(x_2)\cdots q_0(x_K)}\left[ \log \frac{r_\theta(x_1)}{\frac{1}{K}\sum_{z=1}^{K} r_\theta(x_z)} \right].$$

As we alluded to earlier, if we plug-in $p(x|y)$ and $p(x)$ in place of $q_1(x)$ and $q_0(x)$, respectively, then $\mathbb{E}_{p(y)}[\mathcal{L}_{\mathrm{InfoNCE}}(\theta)]$ recovers the standard InfoNCE objective for two modalities.

The following statement from (van den Oord et al., 2018; Poole et al., 2019) is a widely known connection between the InfoNCE objective to the KL divergence, which provides a justification of the InfoNCE objective as an MI estimator for $K$ sufficiently large. We present its proof in Appendix G.1 for completeness.

**Proposition 1.** $\mathcal{D}_{InfoNCE}(\theta) \leq \min\{\log K, D(q_1 \parallel q_0)\}$.

Our first contribution is to provide a *tight* upper bound on $\mathcal{D}_{\mathrm{InfoNCE}}(\theta)$, which yields a much sharper bound on $\mathcal{D}_{\mathrm{InfoNCE}}(\theta)$ than Proposition 1 as a corollary.

**Theorem 2.** *For $z \in [K]$, define $p(x_{1:K}|z)$ as $p(x_{1:K}|z) \triangleq q_1(x_z) \prod_{i \neq z} q_0(x_i)$. Then, we have*

$$\mathcal{D}_{InfoNCE}(\theta) \leq D_{K\text{-}JS}(q_1, q_0) \triangleq \frac{1}{K}\sum_{z=1}^{K} D\Big( p(x_{1:K}|z) \,\Big\|\, \frac{1}{K}\sum_{z'=1}^{K} p(x_{1:K}|z') \Big)$$

$$\leq \min\Big\{ \log K, D(q_1 \parallel q_0) - \log\Big( \frac{1}{K}(2^{D(q_1 \parallel q_0)} - 1) + 1 \Big) \Big\}.$$

*The first inequality is tight if and only if $r_\theta(x) \propto \frac{q_1(x)}{q_0(x)}$.*

This theorem establishes two key theoretical properties of the InfoNCE objective. **First**, the InfoNCE objective is a *tight* variational lower bound of $D_{K\text{-JS}}(q_1, q_0)$, a generalization of Jensen–Shannon divergence (JSD) which we call the *K-way JSD*. The InfoNCE objective becomes equal to the divergence $D_{K\text{-JS}}(q_1, q_0)$ if and only if $r_\theta(x) \propto q_1(x)/q_0(x)$. Since it only learns the density ratio up to a multiplicative constant, one *cannot* use it for a plug-in estimator (i.e., Type 3 in Section 2.1) with the critic (i.e., the density ratio model) learned by InfoNCE. **Second**, the InfoNCE objective may be still far away from $D(q_1 \parallel q_0)$ even for $K$ such that $\log K \geq D(q_1 \parallel q_0)$. Concretely, suppose $D(q_1 \parallel q_0) = 2$. Then, the $\mathcal{D}_{\text{InfoNCE}}(\theta) \leq 1.19\ldots$ when $K = 4$ even if $\log K \geq D(q_1 \parallel q_0)$, and even for $K = 64$, we have $\mathcal{D}_{\text{InfoNCE}}(\theta) \leq 1.93\ldots$, which is strictly smaller than the KL divergence $D(q_1 \parallel q_0) = 2$.

This clearly demonstrates that the InfoNCE objective $\mathcal{D}_{\text{InfoNCE}}(\theta)$ can never match the KL divergence for any finite $K$, even when $\log K \geq D(q_1 \parallel q_0)$. Therefore, InfoNCE is not a variational representation of the KL divergence. This contrasts with other Type 1 estimators such as DV and NWJ, whose bounds become tight when the critic matches the true log-density ratio.

In the next section, we propose a modification of the InfoNCE objective, such that the critic is learned to exactly estimate the density ratio $\frac{q_1(x)}{q_0(x)}$, and so that it can be used in a plug-in estimator for density ratio functionals such as mutual information.

## 3 TENSORIZED DENSITY RATIO ESTIMATION WITH ANCHOR

Consider two distributions $q_0(x)$ and $q_1(x)$. To estimate the density ratio $\frac{q_1(x)}{q_0(x)}$ using samples from $q_0(x)$ and $q_1(x)$, we consider the following classification problem over $\mathcal{X}^K$ (hence *tensorization*), where we define the class densities $p(x_{1:K}|z)$ for $z = 0, 1, \ldots, K$ as

$$
\begin{aligned}
\boxed{\text{class 0 (anchor)}: \ q_0(x_1)q_0(x_2)\cdots q_0(x_K)} \\
\text{class 1}: \ q_1(x_1)q_0(x_2)\cdots q_0(x_K) \\
\text{class 2}: \ q_0(x_1)q_1(x_2)\cdots q_0(x_K) \\
\vdots \\
\text{class } K: \ q_0(x_1)q_0(x_2)\cdots q_1(x_K)
\end{aligned}
\tag{1}
$$

and the prior probabilities over the classes as $p(z) = \frac{\nu}{K+\nu}$ for $z = 0$, and $p(z) = \frac{1}{K+\nu}$ if $z \in [K]$, for some $\nu \geq 0$. As highlighted, class 0 plays a special role as an *anchor*, allowing us to estimate the density ratio without multiplicative ambiguity as long as $\nu > 0$. By *anchor*, we mean that class 0 acts as a fixed reference distribution, eliminating arbitrary scaling and ensuring *identifiability*, which will become precise in Theorem 3 below. We can take $\nu = 0$ if $K \geq 2$ (recovering InfoNCE), but require $\nu > 0$ in the $K = 1$ case to avoid degeneracy. More succinctly, we can write, for $z \neq 0$,

$$
p(x_{1:K}|z) = \frac{q_1(x_z)}{q_0(x_z)} q_0(x_1)q_0(x_2)\cdots q_0(x_K) = \frac{q_1(x_z)}{q_0(x_z)} p(x_{1:K}|z=0).
$$

In words, for $z \neq 0$, the class density is designed such that $x_z$ is drawn from $q_1$, and the rest are from $q_0$. We can write the marginal distribution over $x_{1:K}$ as

$$
p(x_{1:K}) = q_0(x_1)q_0(x_2)\cdots q_0(x_K)\left(\frac{1}{K+\nu}\sum_{i=1}^K \frac{q_1(x_i)}{q_0(x_i)} + \frac{\nu}{K+\nu}\right).
$$

By Bayes' rule, the posterior probability $p(z|x_{1:K})$ is

$$
p(z|x_{1:K}) = \frac{p(x_{1:K}|z)p(z)}{p(x_{1:K})} =
\begin{cases}
\dfrac{\nu}{\nu + \sum_{i=1}^K \frac{q_1(x_i)}{q_0(x_i)}} & \text{if } z = 0 \\[4mm]
\dfrac{\frac{q_1(x_z)}{q_0(x_z)}}{\nu + \sum_{i=1}^K \frac{q_1(x_i)}{q_0(x_i)}} & \text{if } z \in [K]
\end{cases}
\tag{2}
$$

This motivates us to parameterize our probabilistic classifier $p_\theta(z|x_{1:K})$ in the form of

$$p_\theta(z|x_{1:K}) = \begin{cases} \dfrac{\nu}{\nu + \sum_{i=1}^{K} r_\theta(x_i)} & \text{if } z = 0 \\ \dfrac{r_\theta(x_z)}{\nu + \sum_{i=1}^{K} r_\theta(x_i)} & \text{if } z \in [K] \end{cases} . \tag{3}$$

Applying the maximum likelihood estimation (MLE) principle, we can derive the population objective

$$\mathcal{L}_{K;\nu}(\theta) \triangleq -\frac{K}{K+\nu} \mathbb{E}_{q_1(x_1)q_0(x_2)\cdots q_0(x_K)} \left[ \log \frac{r_\theta(x_1)}{\nu + \sum_{i=1}^{K} r_\theta(x_i)} \right]$$

$$- \frac{\nu}{K+\nu} \mathbb{E}_{q_0(x_1)q_0(x_2)\cdots q_0(x_K)} \left[ \log \frac{\nu}{\nu + \sum_{i=1}^{K} r_\theta(x_i)} \right], \tag{4}$$

since $\max_\theta \mathbb{E}_{p(z)p(x_{1:K}|z)}[\log p_\theta(z|x_{1:K})] = \min_\theta \mathcal{L}_{K;\nu}(\theta)$. We call it the *InfoNCE-anchor* objective. Suggested by the name, when $K \geq 2$ and $\nu = 0$, it boils down to InfoNCE. In another extreme, when $K = 1$ and $\nu = 1$, it becomes equivalent to the standard variational lower bound of Jensen–Shannon divergence (see Table 1). In the language of *noise contrastive estimation*, this provides a unification of the standard NCE (Gutmann & Hyvärinen, 2012) ($K = 1, \nu > 0$), and the so-called *ranking NCE* objectives (Ma & Collins, 2018) ($K = 2, \nu = 0$).

**Fisher Consistency.** When $\nu > 0$, it readily follows from the MLE principle that InfoNCE-anchor characterizes the density ratio $\frac{q_1(x)}{q_0(x)}$ as its global minimizer in the population and nonparametric limit.

**Theorem 3** (Fisher consistency). *Let $\theta^* \triangleq \arg\min_\theta \mathcal{L}_{K;\nu}(\theta)$ denote a global optimizer of the InfoNCE-anchor objective. Suppose that there exists $\theta_0$ such that $r_{\theta_0}(x) = \frac{q_1(x)}{q_0(x)}$. If $K \geq 1$ and $\nu > 0$, $r_{\theta^*}(x) = \frac{q_1(x)}{q_0(x)}$ for almost every $x$ under $q_0$. If $K \geq 2$ with $\nu = 0$, there exists some constant $C > 0$ such that $r_{\theta^*}(x) = C\frac{q_1(x)}{q_0(x)}$ for $q_0$-almost every $x$.*

### 3.1 APPLICATION: DIVERGENCE ESTIMATION AND REPRESENTATION LEARNING

We can apply the InfoNCE-anchor objective to estimate MI or to learn representation when given a joint distribution $p(x, y)$, in a similar way to InfoNCE (van den Oord et al., 2018). That is, for each $y$, we can apply the InfoNCE-anchor for $q_1(x) \leftarrow p(x|y)$ and $q_0(x) \leftarrow p(x)$.[1] For the final objective, we take an expectation over $y \sim p(y)$:

$$\mathcal{L}_{K;\nu}^{(1)}(\theta) \triangleq \mathbb{E}_{p(y)} \left[ -\frac{K}{K+\nu} \mathbb{E}_{p(x_1|y)p(x_2)\cdots p(x_K)} \left[ \log \frac{r_\theta(x_1, y)}{\nu + \sum_{i=1}^{K} r_\theta(x_i, y)} \right] \right.$$

$$\left. - \frac{\nu}{K+\nu} \mathbb{E}_{p(x_1)p(x_2)\cdots p(x_K)} \left[ \log \frac{\nu}{\nu + \sum_{i=1}^{K} r_\theta(x_i, y)} \right] \right].$$

When $\nu = 0$ with $K \geq 2$, it boils down to the original InfoNCE, and minimizing $\mathcal{L}_{K;0}^{(1)}(\theta)$ can only guarantee that for some function $C(y)$, $r_{\theta^*}(x, y) = C(y)\frac{p(x,y)}{p(x)p(y)}$. When applied to representation learning, the vanilla InfoNCE (i.e., with $\nu = 0$) thus may lead to an undesirable behavior due to uncontrollable $C(y)$, whereas the anchor (i.e., $\nu > 0$) can remove such degeneracy. However, in our representation learning experiment, we observe that the anchor does not lead to the improvement of downstream task performance; see Section 4.3.

With a minibatch of size $B$, we can implement the loss with anchor for $K = B - 1$ as follows:

$$-\frac{K}{K+\nu} \frac{1}{B} \sum_{b=1}^{B} \log \frac{r_{bb}}{\nu + \sum_{j\in[B]\setminus\{b-1\}} r_{bj}} - \frac{\nu}{K+\nu} \frac{1}{B} \sum_{b=1}^{B} \log \frac{\nu}{\nu + \sum_{j\in[B]\setminus\{b\}} r_{bj}}.$$

We note that the computational overhead introduced by the additional second term, compared to InfoNCE, is negligible. We provide further discussion along with pseudocode in Appendix C.

---

[1] An alternative approach is to set $(q_1(x), q_0(x)) \leftarrow (p(x, y), p(x)p(y))$; see Appendix B.

The density ratio estimator is typically parameterized as $r_\theta(x,y) \leftarrow e^{c_\theta(x,y)}$, where $c_\theta(x,y)$ (the *critic*) is often a neural network. In representation learning, common choices are the exponential form $r_\theta(x,y) \leftarrow e^{\frac{1}{\tau}\frac{\mathbf{f}_\theta(x)^\top \mathbf{g}_\theta(y)}{\|\mathbf{f}_\theta(x)\|_2\|\mathbf{g}_\theta(y)\|_2}}$ (see, e.g., (van den Oord et al., 2018)) or the direct form $r_\theta(x,y) \leftarrow \frac{1}{\tau}\frac{\mathbf{f}_\theta(x)^\top \mathbf{g}_\theta(y)}{\|\mathbf{f}_\theta(x)\|_2\|\mathbf{g}_\theta(y)\|_2}$ (see, e.g., (HaoChen et al., 2021)), such that $\mathbf{f}_\theta(x)$ and $\mathbf{g}_\theta(y)$ are learned embeddings that approximate PMI or PD, respectively. Here, $\tau > 0$ is a *temperature* parameter.

## 3.2 INFONCE-ANCHOR INTERPOLATES DV AND NWJ BOUNDS WHEN $K \to \infty$

One may ask about the behavior of InfoNCE-anchor when $K \to \infty$. While we defer a rigorous statement (Theorem 10) to Appendix D, we remark that InfoNCE-anchor, by setting $\nu$ to vary as $K \to \infty$ such that $\nu/K \to \beta$ for some $\beta \geq 0$, we can show that InfoNCE-anchor behaves similar to a generalization of the DV bound, which can be rearranged to yield

$$\mathbb{E}_{q_1(x)}[\log r_\theta(x)] - (\beta+1)\log\left(\frac{\beta}{\beta+1} + \frac{1}{\beta+1}\mathbb{E}_{q_0(x)}[r_\theta(x)]\right) \leq D(q_1 \| q_0). \tag{5}$$

When $\beta = 0$, this boils down to the standard DV bound. When $\beta \to \infty$, the left-hand side becomes $\mathbb{E}_{q_1(x)}[\log r_\theta(x)] - \mathbb{E}_{q_0(x)}[r_\theta(x)] + 1 \leq D(q_1 \| q_0)$, which is the NWJ bound. Moreover, we can even show that this bound *monotonically* interpolates between the DV bound (tightest, $\beta = 0$) and the NWJ bound (loosest, $\beta = \infty$). A similar asymptotic behavior of InfoNCE (i.e., for $\nu = 0$) was noted by Wang & Isola (2020), but specifically in the context of contrastive representation learning.

## 3.3 DISCUSSION ON EXISTING VARIANTS OF INFONCE ESTIMATOR

In this section, we discuss two existing variants of InfoNCE, which were proposed in the effort of fixing the aforementioned issues of InfoNCE as the MI estimator. We highlight why they are insufficient as a fundamental fix, and how different from our proposal.

$\alpha$**-InfoNCE.** Poole et al. (2019) proposed an alternative estimator called $\alpha$-InfoNCE, defined as

$$\mathcal{D}_{\alpha\text{-InfoNCE}}(\theta) \triangleq \mathbb{E}_{q_1(x_1)q_0(x_2)\cdots q_0(x_K)}\left[\log \frac{r_\theta(x_1)}{\alpha r_\theta(x_1) + \frac{1-\alpha}{K-1}\sum_{z=2}^{K} r_\theta(x_z)}\right]$$

for some $\alpha \in (0, \frac{1}{K}]$. Note that setting $\alpha \leftarrow \frac{1}{K}$ recovers the original InfoNCE bound. For $\alpha < \frac{1}{K}$, this quantity can neither be understood as a loss for classification nor be a variational lower bound for $D(q_1 \| q_0)$. Lee & Shin (2022, Theorem 4.2) claimed that the $\alpha$-InfoNCE objective is a *tight* variational lower bound for a $\alpha$-skew KL divergence $D(q_1 \| \alpha q_1 + (1-\alpha)q_0)$, that is, $\mathcal{D}_{\alpha\text{-InfoNCE}}(\theta) \leq D(q_1 \| \alpha q_1 + (1-\alpha)q_0)$ for $\alpha \in [0, \frac{1}{2}]$, and the equality can be achieved. We find, however, the proof has a flaw and we can indeed show that, unless $q_1 = q_0$, we have

$$\sup_\theta \mathcal{D}_{\alpha\text{-InfoNCE}}(\theta) > D(q_1 \| \alpha q_1 + (1-\alpha)q_0) \tag{6}$$

in the nonparametric limit. We defer the flaw in the proof, and the proof of Eq. (6) to Appendix A.

**MLInfoNCE.** Song & Ermon (2020b) introduced the *multi-label InfoNCE (MLInfoNCE)* estimator defined as

$$\mathcal{D}_{\text{MLInfoNCE}}(\theta) \triangleq \mathbb{E}_{\prod_{w=1}^{m} q_1(x_{w1})\prod_{z=2}^{k} q_0(x_{wz})}\left[\sum_{w=1}^{m} \log \frac{r_\theta(x_{w1})}{\sum_{w'=1}^{m}(r_\theta(x_{w'1}) + \sum_{z=2}^{k} r_\theta(x_{w'z}))}\right]. \tag{7}$$

Song & Ermon (2020b, Theorem 2) shows that $\mathcal{D}_{\text{MLInfoNCE}}(\theta) \leq D(q_1 \| q_0)$. However, we note that this objective cannot be understood as a loss derived from a proper classification setup unlike InfoNCE-anchor.

## 3.4 EXTENSION WITH PROPER SCORING RULES

In the classification setup of Eq. (1), density ratio estimation reduces to estimating the class probability $p(z|x_{1:K})$ in Eq. (2) via the model $p_\theta(z|x_{1:K})$ in Eq. (3). The cross-entropy loss in Eq. (4) is a

*proper* scoring rule, ensuring that the optimized model recovers the true posterior. More generally, once density ratio estimation is cast as class probability estimation, *any proper scoring rule* can be applied, yielding a broad family of consistent objectives.

Here we start with a general description of the proper scoring rules (Gneiting & Raftery, 2007; Dawid et al., 2012). Let $\mathcal{Z}$ be a discrete alphabet and let $\mathcal{A}$ be any alphabet. Suppose that we have sample access to the underlying distribution $p(a, z)$ over $\mathcal{A} \times \mathcal{Z}$. The goal of class probability estimation (CPE) (Garcia-Garcia & Williamson, 2012) is to estimate the underlying class probability $\boldsymbol{\eta} \colon \mathcal{A} \to \Delta(\mathcal{Z})$, where $\boldsymbol{\eta}(a) \triangleq (p(z|a))_{z \in \mathcal{Z}}$, using samples from $p(a, z)$.

To characterize a class probability estimator as the optimizer of an optimization problem, we consider a tuple of loss functions $\boldsymbol{\lambda} = (\lambda_z \colon \Delta(\mathcal{Z}) \to \mathbb{R})_{z \in \mathcal{Z}}$, which we call a *scoring rule*, whereby an *action* $\hat{\boldsymbol{\eta}} \colon \mathcal{A} \to \Delta(\mathcal{Z})$ incurs loss $\lambda_z(\hat{\boldsymbol{\eta}}(a))$ for a data point $(a, z)$. Then, we measure the performance of an action $\hat{\boldsymbol{\eta}}$ by the expected loss $\mathbb{E}_{p(a,z)}[\lambda_z(\hat{\boldsymbol{\eta}}(a))]$.

**Definition 4** (Proper scoring rules). A scoring rule $\boldsymbol{\lambda} \colon \Delta(\mathcal{Z}) \to \mathbb{R}^{\mathcal{Z}}$ is a vector-valued loss function. A scoring rule is said to be *proper* if $\boldsymbol{\eta}$ is optimal with respect to $\boldsymbol{\lambda}$, i.e., for any distribution $p(a, z)$,

$$\boldsymbol{\eta}(\cdot) \in \arg\min_{\hat{\eta} \colon \mathcal{A} \to \Delta(\mathcal{Z})} \mathbb{E}_{p(a,z)}[\lambda_z(\hat{\boldsymbol{\eta}}(a))].$$

If $\boldsymbol{\eta}$ is the *unique* optimal solution with respect to $\boldsymbol{\lambda}$, then $\boldsymbol{\lambda}$ is said to be *strictly* proper.

We note that most (strictly) proper scoring rules can be induced by a (strictly) differentiable convex function. For the sake of exposition, let $\mathcal{Z} = \{0, \dots, M\}$ concretely. Then, for a differentiable function $\Psi \colon \{1\} \times \mathbb{R}_+^M \to \mathbb{R}$, we define the $\Psi$-induced scoring rule as

$$\boldsymbol{\lambda}^\Psi(\boldsymbol{\eta}) \triangleq \left[ \begin{array}{c} \langle \boldsymbol{\rho}, \nabla_{\boldsymbol{\rho}} \Psi(\boldsymbol{\rho}) \rangle - \Psi(\boldsymbol{\rho}) \\ (-\nabla_{\boldsymbol{\rho}} \Psi(\boldsymbol{\rho}))_{1:M} \end{array} \right] \Bigg|_{\boldsymbol{\rho} = (1, \frac{\eta_1}{\eta_0}, \dots, \frac{\eta_M}{\eta_0})}. \tag{8}$$

**Proposition 5.** *If $\Psi$ is (strictly) convex and twice differentiable, $\boldsymbol{\lambda}^\Psi$ is (strictly) proper.*

The canonical example is the log score, which results in InfoNCE-anchor in Eq. (4). We present some examples of proper scoring rules and the generating convex functions in Appendix E.5.

Now, considering the classification setup in Eq. (1), let $\boldsymbol{\lambda} = \boldsymbol{\lambda}^\Psi$ be a strictly proper scoring rule over discrete alphabet $\mathcal{Z} = \{0, \dots, K\}$, induced by a strictly convex function $\Psi \colon \mathbb{R}_+^K \to \mathbb{R}$. Applying the scoring rule to evaluate the score of the class probability $p_\theta(z|x_{1:K})$ (in Eq. (3)) with respect to the underlying distribution $p(z)p(x_{1:K}|z)$, we can write the population objective (to be minimized) as

$$\mathcal{L}_{K;\nu}^\Psi(\boldsymbol{\eta}_\theta) \triangleq \mathbb{E}_{p(x_{1:K}, z)}[\lambda_z(\boldsymbol{\eta}_\theta(x_{1:K}))], \tag{9}$$

where we use $\boldsymbol{\eta}_\theta(x_{1:K}) = (p_\theta(z|x_{1:K}))_{z \in \mathcal{Z}}$ to denote the class probability vector. Let $\boldsymbol{\eta}^*(x_{1:K})$ denote the underlying class probability $(p(z|x_{1:K}))_{z \in \mathcal{Z}}$. The following statement subsumes Theorem 3 as a special case. In what follows, for a differentiable, convex function $h \colon \mathbb{R}^d \to \mathbb{R}$, we define the *Bregman divergence* between $\mathbf{x}, \mathbf{y} \in \mathbb{R}^d$ generated by $h$ as $B_h(\mathbf{x}, \mathbf{y}) \triangleq h(\mathbf{x}) - h(\mathbf{y}) - \nabla h(\mathbf{y})^\intercal (\mathbf{x} - \mathbf{y})$.

**Theorem 6.** *For $\nu > 0$,*

$$\mathcal{L}_{K;\nu}^\Psi(\boldsymbol{\eta}_\theta) - \mathcal{L}_{K;\nu}^\Psi(\boldsymbol{\eta}^*) = \frac{\nu}{K + \nu} \mathbb{E}_{q_0(x_1)q_0(x_2)\cdots q_0(x_K)} \left[ B_\Psi \left( \frac{\mathbf{r}^*(x_{1:K})}{\nu}, \frac{\mathbf{r}_\theta(x_{1:K})}{\nu} \right) \right],$$

*where $\mathbf{r}^*(x_{1:K}) \triangleq \left( \frac{q_1(x_z)}{q_0(x_z)} \right)_{z \in [K]}$ and $\mathbf{r}_\theta(x_{1:K}) \triangleq (r_\theta(x_z))_{z \in [K]}$. If $\Psi$ is convex, we have*

$$-\mathcal{L}_{K;\nu}^\Psi(\boldsymbol{\eta}_\theta) \le -\mathcal{L}_{K;\nu}^\Psi(\boldsymbol{\eta}^*) = \frac{\nu}{K + \nu} \mathbb{E}_{q_0(x_1)q_0(x_2)\cdots q_0(x_K)} \left[ \Psi \left( \frac{\mathbf{r}^*(x_{1:K})}{\nu} \right) \right]. \tag{10}$$

*If $\Psi$ is (strictly) convex, the equality is achieved if (and only if) $r_\theta(x) = \frac{q_1(x)}{q_0(x)}$.*

That is, in principle, we can minimize the DRE objective function in Eq. (9) for any strongly convex, differentiable function $\Psi$ to estimate the density ratio. Beyond the consistency, as stated in Eq. (10), this corollary also shows that the DRE objective (with negation) can be understood as a variational lower bound of some divergence between $q_1(x)$ and $q_0(x)$ induced by $\Psi$, defined as

$\mathbb{E}_{q_0(x_1)q_0(x_2)\cdots q_0(x_K)}[\Psi(\frac{\mathbf{r}^*(x_{1:K})}{\nu})]$. This is analogous to that the InfoNCE-anchor objective in Eq. (4) is a variational lower bound of the $K$-way JSD $D_{K\text{-JS}}(q_1, q_0)$. We note that this extension can be viewed as a special application of the more general multi-distribution density ratio estimation studied by Yu et al. (2021), for the binary density ratio estimation.

**Implementation.** Similar to InfoNCE-anchor in Eq. (4), this objective function can be simplified further if the scoring rule satisfies a mild symmetry condition; see Appendix F.2.

**Alternative Characterization of Proper Scoring Rule.** One minor limitation of the characterization in Theorem 6 is that $\nu = 0$ is not permitted as a special case, and thus InfoNCE cannot be subsumed. In Appendix F.1, we provide an alternative characterization of proper scoring rules, which can be related to the above formulation via the *perspective transformation*, and admits $\nu = 0$.

**Special Cases.** For the special case when $K = 1$ and $\nu = 1$, note that the right hand side becomes the $f$-divergence $D_f(q_1 \parallel q_0)$ when $f = \Psi$. That is, the objective boils down to the standard variational lower bound on the $f$-divergence (Nguyen et al., 2010), hence recovering the $f$-DIME objectives of Letizia et al. (2024) and $f$-MICL objectives of Lu et al. (2023). We also note that the GAN-DIME and HD-DIME estimators in (Letizia et al., 2024) are essentially identical to the estimators proposed in (Tsai et al., 2020). More examples can be found in Appendix F.3.

## 4 EXPERIMENTS

In this section, we show that InfoNCE-anchor outperforms existing estimators in MI estimation (Section 4.1) and downstream classification task (Section 4.2). We also report a negative result: anchor variants do not improve the representation quality of InfoNCE in self-supervised representation learning tasks (Section 4.3). In all experiments, we set $K = B - 1$ and use $\nu = 1$ as the default. Notably, this default setting consistently outperforms the baseline objectives without any hyperparameter tuning. We provide a detailed sensitivity analysis in Appendix H.4.

### 4.1 MI ESTIMATION

We evaluate various neural MI estimators on structured and unstructured data using the benchmark suite of Lee & Rhee (2024).Experiments cover three domains: multivariate Gaussian data, MNIST images, and BERT embeddings of IMDB reviews. To control ground truth MI, the benchmark employs same-class sampling for positive pairs and a binary symmetric channel to inject controlled noise. This allows systematic variation of MI from 2 to 10 bits in 2-bit increments. All experiments were done with batch size 64 and averaged over 20 random runs. Implementation details such as critic architectures and optimization setups are deferred to Appendix H.1.

Figure 1 summarizes the results. We emphasize that InfoNCE-anchor achieves both lower bias and comparable variance across domains. $\mathsf{JS}_{\text{plugin}}$ (equivalent to InfoNCE-anchor with $K = 1$, $\nu = 1$) performs comparably on Gaussians but deteriorates on higher-dimensional tasks such as MNIST and texts, highlighting the value of large $K$. We also evaluate Spherical, an InfoNCE-anchor variant induced by the spherical scoring rule (Gneiting & Raftery, 2007); see Table 6 in Appendix F.3. Its inferior performance indicates that, despite the equivalence of strictly proper scoring rules, the log score remains the most effective in practice. Additional results for Gaussian with varying batch sizes can be found in Appendix H.1.

### 4.2 PROTEIN INTERACTION PREDICTION

As a further demonstration of the effectiveness of InfoNCE-anchor, we perform an experiment from a recent study by Gowri et al. (2024). In the work, the authors examined protein embeddings derived from a pretrained protein language model (pLM), the *ProtTrans5* model (Elnaggar et al., 2021), and evaluated whether one can predict interactions between protein pairs $(x, y)$, specifically, $(K, T) =$ (kinase, target) and $(L, R) =$ (ligand, receptor) pairs in the considered setting. The interaction labels are from the *OmniPath* database (Türei et al., 2021). We ran the experiment following the same setup, with estimating the PMI using the JS, InfoNCE-anchor, and a few other variational approaches, and using them to decide whether interaction exists by thresholding the PMI of a given pair.

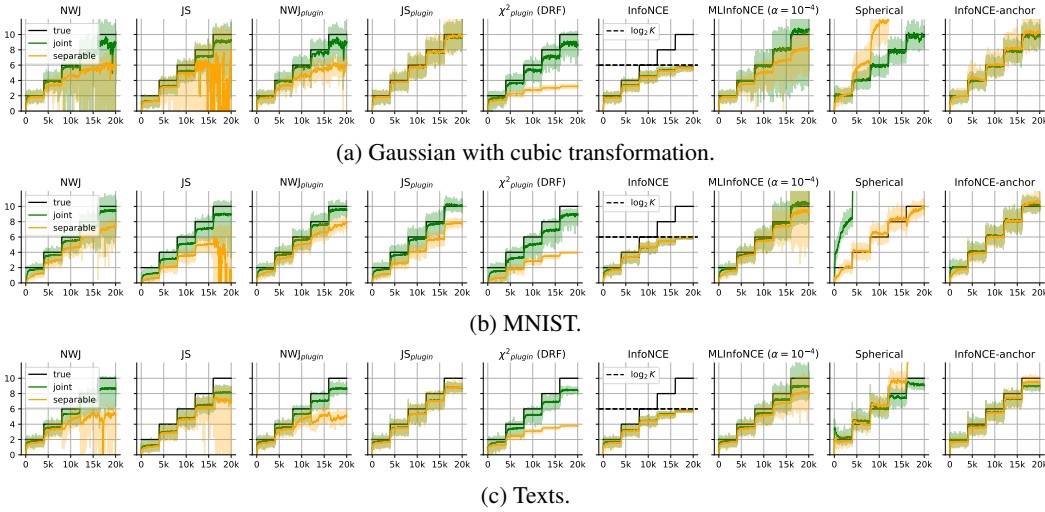

(a) Gaussian with cubic transformation.

(b) MNIST.

(c) Texts.

Figure 1: Summary of MI estimation results on the standard benchmark. Across all the cases, the proposed InfoNCE-anchor estimator (the rightmost column) consistently demonstrates low-bias, low-variance performance compared to the existing estimators. We note that $JS_{plugin}$ is equivalent to JSD-LB (Dorent et al., 2025).

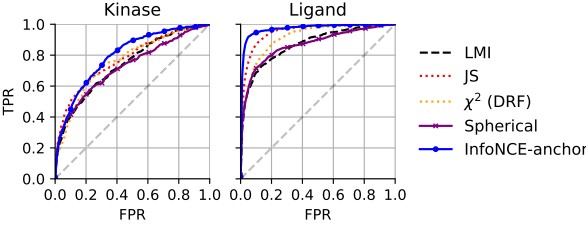

(b) Summary of AUROC performance on the Kinase and Ligand benchmarks (mean$_{\pm std}$ computed over 20 runs).

| Objective | Kinase | Ligand |
|---|---|---|
| LMI | $0.74_{\pm 0.08}$ | $0.87_{\pm 0.04}$ |
| $\chi^2$ (DRF) | $0.76_{\pm 0.07}$ | $0.92_{\pm 0.03}$ |
| JS | $0.77_{\pm 0.08}$ | $0.95_{\pm 0.02}$ |
| InfoNCE-anchor | $\mathbf{0.80}_{\pm \mathbf{0.06}}$ | $\mathbf{0.97}_{\pm \mathbf{0.01}}$ |
| Spherical | $0.73_{\pm 0.07}$ | $0.87_{\pm 0.05}$ |

(a) ROC curves for each problem instance.

Figure 2: Summary of the protein interaction prediction experiment.

Figure 2a and Figure 2b summarize the results. As shown in Figure 2a, InfoNCE-anchor shows the best prediction results for both problem instances, while the JS estimator, which is a special case of InfoNCE-anchor when $K = 1$ and $\nu = 1$, performs second best. This again demonstrates the practical benefit of large $K$ for accurate density ratio estimation. We also recall that the standard InfoNCE objective is not suitable in this scenario, as it only estimates PMI up to a multiplication with an arbitrary function $C(y)$ discussed in Section 3.1. We include the histograms of learned PMI values (Figure 5) as well as the ROC curves of each estimator for different runs (Figure 4) in Appendix H.2.

## 4.3 SELF-SUPERVISED REPRESENTATION LEARNING

In earlier sections we showed that InfoNCE-anchor improves MI estimation and downstream tasks using the learned density ratio model. A natural question is whether this benefit carries over to self-supervised learning (SSL), where InfoNCE is the standard objective. We therefore pretrain a ResNet-18 on CIFAR-100 using the `solo-learn` framework (da Costa et al., 2022), comparing several contrastive objectives under identical settings (batch size $B = 256$, same optimizer), and evaluate representations via linear probing; see Appendix H.3 for results with different backbones.

Table 2 shows that InfoNCE continues to yield the strongest representations. Adding the anchor, despite improving density ratio estimation, does not translate into better SSL performance. This suggests that the uncontrollable multiplicative factor $C(y)$ in InfoNCE is either nearly constant or irrelevant for representation learning. JS performs poorly, highlighting the importance of large $K$, while spherical scores collapse entirely, likely due to unfavorable optimization dynamics.

Table 2: Linear probing accuracy (%) after SSL pretraining. We used PD parameterization for Spherical and $\chi^2$. Detailed setups can be found in Appendix H.3.

| Objective | InfoNCE | InfoNCE-anchor | Spherical | JS | $\chi^2$ |
|---|---|---|---|---|---|
| Top-1 accuracy | 65.98 | 65.74 | 4.33 | 61.69 | 65.59 |
| Top-5 accuracy | 89.69 | 89.24 | 17.91 | 87.33 | 88.4 |

**On the Failure of Other Proper Scoring Rules.** We highlighted the spherical scoring rule because it is a concrete example from the InfoNCE-anchor family (see Table 6) that consistently fails in practice. Among these objectives, the log scoring rule is the only instance that reliably works. As noted in Theorem 6, all scoring rules are Fisher-consistent in the nonparametric and population limits; thus, their differences are not due to statistical inconsistency. Instead, the empirically observed discrepancies arise from the optimization landscape.

For the scoring rules that fail, we observe that the training objectives plateau almost immediately and do not improve, indicating that the optimization geometry induced by these objectives is unfavorable for representation learning at realistic batch sizes, model parameterizations, and noise levels. While understanding the precise landscape properties that distinguish successful and unsuccessful instances is an interesting direction, a full characterization is beyond the scope of this work.

**Structural Analysis of Learned Representations.** To understand why the improved MI estimation of InfoNCE-anchor does not translate to gains in self-supervised representation learning, we analyze the structural properties of the learned features. We compare the representations learned by standard InfoNCE and InfoNCE-anchor using Centered Kernel Alignment (CKA) (Kornblith et al., 2019) and the Uniformity and Alignment metrics (Wang & Isola, 2020). We observe high similarity between the two representations (CKA $\approx 0.88$ for both the backbone and projector), suggesting that the anchor does not fundamentally alter the global feature geometry. The marginal improvements in Uniformity (0.350 vs. 0.357) are balanced by a slight degradation in Alignment (-3.77 vs. -3.81). This suggests that while the anchor modification encourages a more distributed feature space, it may slightly relax the local clustering of positive pairs, resulting in the observed parity in classification accuracy.

These findings suggest a decoupling between MI estimation accuracy and representation quality. While the anchor class resolves the density-ratio identifiability issue necessary for accurate MI, standard InfoNCE already provides a sufficient training signal for learning separable features. This supports our conclusion that contrastive learning benefits primarily from the learning of structured density ratios rather than the absolute scale of mutual information.

Overall, these findings indicate that neither accurate MI estimation nor exact density ratio recovery is essential for high-quality representation. Our results suggest that what matters in SSL appears to be the factorization of PMI, benefit of large $K$, and favorable optimization properties with the log score.

## 5 Concluding Remarks

We revisited InfoNCE and showed it is not a consistent MI estimator but a variational bound of some other divergence. We introduce InfoNCE-anchor, a simple fix enabling consistent density ratio estimation within a unified scoring-rule framework. InfoNCE-anchor sets new state-of-the-art MI estimation benchmarks and aids predictive tasks, though it does not improve SSL performance, highlighting that accurately estimating MI is not essential for representation quality (Tschannen et al., 2020). We hope our work clarifies the role of InfoNCE and MI estimation in contrastive learning.

### Acknowledgments

This work was supported, in part, by the MIT-IBM Watson AI Lab under Agreement No. W1771646, and by the United States Air Force Research Laboratory and the United States Air Force Artificial Intelligence Accelerator under Cooperative Agreement Number FA8750-19-2-1000. The views and conclusions contained in this document are those of the authors and should not be interpreted as representing the official policies, either expressed or implied, of the Department of the Air Force or the U.S. Government. The U.S. Government is authorized to reproduce and distribute reprints for Government purposes notwithstanding any copyright notation herein.

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

APPENDIX

## A   CLARIFICATION ON $\alpha$-INFONCE

In Section 3.3, we noted that Lee & Shin (2022, Theorem 4.2) is not valid as stated in the nonparametric limit. Here, we present a corrected statement and explain why the argument in their proof does not establish the claimed identity.

We first restate their statement.

**Theorem 7** (Lee & Shin, 2022, Theorem 4.2). *Let $\alpha \in (0, \frac{1}{2}]$. In the nonparametric limit,*

$$\sup_{\theta} \mathcal{D}_{\alpha\text{-InfoNCE}}(\theta) = D(q_1 \parallel \alpha q_1 + (1 - \alpha)q_0).$$

They also state an analogous result for the $\alpha$-skewed version of MLInfoNCE; for brevity, we focus on $\alpha$-InfoNCE, since the same issue arises there.

Below, we give the correct characterization, which shows that the equality in Theorem 7 fails in general. We then identify the precise step where the original proof breaks down.

### A.1   CORRECTED STATEMENT

We can prove the following:

**Proposition 8.** *For any $\alpha \in [0, 1]$, in the nonparametric limit,*

$$\sup_{\theta} \mathcal{D}_{\alpha\text{-InfoNCE}}(\theta) \geq D(q_1 \parallel \alpha q_1 + (1 - \alpha)q_0),$$

*where the equality holds if and only if $q_1 = q_0$.*

That is, if $q_1 \neq q_0$, then we have a strict inequality

$$\sup_\theta \mathcal{D}_{\alpha\text{-InfoNCE}}(\theta) > D(q_1 \parallel \alpha q_1 + (1-\alpha)q_0),$$

which invalidates Theorem 7.

We now prove the proposition.

*Proof of Proposition 8.* In the nonparametric limit, we can consider $\theta_o$ which satisfies $r_\theta(x) = \frac{q_1(x)}{q_0(x)}$. Let $p_z(x_{1:K}) \triangleq \frac{q_1(x_z)}{q_0(x_z)} q_0(x_1)q_0(x_2)\cdots q_0(x_K)$ denote the class density for class $z \in [K]$ as defined in Eq. (1). Then, for such $\theta_o$, the $\alpha$-InfoNCE objective can be written as

$$\mathcal{D}_{\alpha\text{-InfoNCE}}(\theta_o) = \mathbb{E}_{p_1(x_{1:K})} \left[ \log \frac{p_1(x_{1:K})}{\alpha p_1(x_{1:K}) + \frac{1-\alpha}{K-1} \sum_{z=2}^K p_z(x_{1:K})} \right]$$

$$= D\left( p_1(x_{1:K}) \,\middle\|\, \alpha p_1(x_{1:K}) + \frac{1-\alpha}{K-1} \sum_{z=2}^K p_z(x_{1:K}) \right).$$

Here, we invoke the monotonicity of the KL divergence (Cover & Thomas, 2006): for any joint distributions $p(x, y)$ and $q(x, y)$, if $p(x)$ and $q(x)$ are the marginal distributions, respectively, we have

$$D(p(x, y) \parallel q(x, y)) = D(p(x) \parallel q(x)) + \mathbb{E}_{p(x)}[D(p(y|x) \parallel q(y|x))]$$
$$\geq D(p(x) \parallel q(x)),$$

where the equality holds if and only if $\mathbb{E}_{p(x)}[D(p(y|x) \parallel q(y|x))] = 0$, or equivalently $p(y|x) = q(y|x)$ for $p(x)$-almost everywhere (a.e.). Since the marginal distribution of the two distributions $p_1(x_{1:K})$ and $\alpha p_1(x_{1:K}) + \frac{1-\alpha}{K-1} \sum_{z=2}^K p_z(x_{1:K})$ over $x_1$ are $q_1(x_1)$ and $q_1(x_1) + \frac{1-\alpha}{K-1} \sum_{z=2}^K q_0(x_1) = \alpha q_1(x_1) + (1-\alpha)q_0(x_1)$, respectively, we have the following chain of inequalities:

$$\sup_\theta \mathcal{D}_{\alpha\text{-InfoNCE}}(\theta) \geq \mathcal{D}_{\alpha\text{-InfoNCE}}(\theta_o)$$

$$= D\left( p_1(x_{1:K}) \,\middle\|\, \alpha p_1(x_{1:K}) + \frac{1-\alpha}{K-1} \sum_{z=2}^K p_z(x_{1:K}) \right)$$

$$\geq D\left( q_1(x_1) \,\middle\|\, \alpha q_1(x_1) + (1-\alpha)q_0(x_1) \right).$$

It remains to characterize the equality condition. From the monotonicity lemma, the last equality holds if and only if the following holds for $q_1(x_1)$-a.e.:

$$\frac{p_1(x_{1:K})}{q_1(x_1)} = \frac{\alpha p_1(x_{1:K}) + \frac{1-\alpha}{K-1} \sum_{z=2}^K p_z(x_{1:K})}{q_0(x_1)}$$

$$\iff \alpha \frac{q_1(x_1)}{q_0(x_1)} + \frac{1-\alpha}{K-1} \sum_{z=2}^K \frac{q_1(x_z)}{q_0(x_z)} = 1.$$

The only way the above equality can hold almost surely is that $q_1(x)/q_0(x)$ is constant almost everywhere, which implies $q_1 = q_0$. $\qquad\square$

### A.2 ISSUE IN THE ORIGINAL PROOF

To prove this statement, Lee & Shin (2022) used the sandwich argument:

$$\mathcal{D}_{DV}(\theta; \alpha q_1 + (1-\alpha)q_0) \overset{(a)}{\leq} \mathcal{D}_{\alpha\text{-InfoNCE}}(\theta) \overset{(b)}{\leq} D(q_1 \parallel \alpha q_1 + (1-\alpha)q_0).$$

If both inequalities held, since the DV bound is the tight lower bound on the KL divergence which becomes tight in the nonparametric limit, we can conclude equality in Theorem 7. However, Proposition 8 states a stronger inequality than $(a)$, and implies that $(b)$ cannot hold.

Since $(a)$ follows as a corollary of Proposition 8, we now detail the specific flaw in their proof for $(b)$. To argue $(b)$, they invoke the NWJ bound. That is,

$$D(p \parallel q) \geq \mathcal{D}_{\mathsf{NWJ}}(\theta; p, q) \triangleq \mathbb{E}_{p(x)}[\log \tilde{r}_\theta(x)] - \mathbb{E}_{q(x)}[\tilde{r}_\theta(x)] + 1,$$

where the equality holds if and only if $\tilde{r}_\theta(x) = \frac{p(x)}{q(x)}$ $q$-almost everywhere. For fixed $x_2, \ldots, x_K$, they plug in a test function

$$\tilde{r}_\theta(x_1) \leftarrow \frac{r_\theta(x_1)}{\alpha r_\theta(x_1) + \frac{1-\alpha}{K-1} \sum_{z=2}^K r_\theta(x_z)}$$

to $\mathcal{D}_{\mathsf{NWJ}}(\theta; q_1, \alpha q_1 + (1-\alpha)q_0)$, which yields

$$D(q_1 \parallel \alpha q_1 + (1-\alpha)q_0) \geq \mathbb{E}_{q_1(x_1)} \left[ \log \frac{r_\theta(x_1)}{\alpha r_\theta(x_1) + \frac{1-\alpha}{K-1} \sum_{z=2}^K r_\theta(x_z)} \right]$$
$$+ 1 - \mathbb{E}_{\alpha q_1(x_1) + (1-\alpha)q_0(x_1)} \left[ \frac{r_\theta(x_1)}{\alpha r_\theta(x_1) + \frac{1-\alpha}{K-1} \sum_{z=2}^K r_\theta(x_z)} \right].$$

Taking expectation over $q_0(x_2) \ldots q_0(x_K)$, we have

$$D(q_1 \parallel \alpha q_1 + (1-\alpha)q_0)$$
$$\geq \mathcal{D}_{\alpha\text{-InfoNCE}}(\theta) + 1 - \mathbb{E}_{q_0(x_2)\ldots q_0(x_K)} \left[ \mathbb{E}_{\alpha q_1(x_1) + (1-\alpha)q_0(x_1)} \left[ \frac{r_\theta(x_1)}{\alpha r_\theta(x_1) + \frac{1-\alpha}{K-1} \sum_{z=2}^K r_\theta(x_z)} \right] \right].$$

To conclude $D(q_1 \parallel \alpha q_1 + (1-\alpha)q_0) \geq \mathcal{D}_{\alpha\text{-InfoNCE}}(\theta)$, then claim that

$$\mathbb{E}_{q_0(x_2)\ldots q_0(x_K)} \left[ \mathbb{E}_{\alpha q_1(x_1) + (1-\alpha)q_0(x_1)} \left[ \frac{r_\theta(x_1)}{\alpha r_\theta(x_1) + \frac{1-\alpha}{K-1} \sum_{z=2}^K r_\theta(x_z)} \right] \right] \overset{(c)}{\leq} 1$$

for $\alpha \in (0, \frac{1}{2}]$. This is argued based on the following propositions from (Song & Ermon, 2020b), which we rephrase for simplicity:

**Proposition 9** (Song & Ermon, 2020b, Propositions 1 and 2). *If $Z_1, Z_2, \ldots, Z_K$ are exchangeable, then*

$$\mathbb{E} \left[ \frac{Z_1}{\alpha Z_1 + \frac{1-\alpha}{K-1} \sum_{j=2}^K Z_j} \right] \leq \begin{cases} \frac{1}{\alpha K} & \text{if } \alpha \in (0, \frac{2}{K}], \\ 1 & \text{if } \alpha \in [\frac{1}{K}, \frac{1}{2}]. \end{cases}$$

That is, Lee & Shin (2022) apply this proposition to argue $(c)$, by letting $(Z_1, \ldots, Z_K) \leftarrow (r_\theta(x_1), \ldots, r_\theta(x_K))$. Note, however, that we cannot apply the proposition here, as the random variables are *not exchangeable* under $(x_1, \ldots, x_K) \sim (\alpha q_1(x_1) + (1-\alpha)q_0(x_1))q_0(x_2) \ldots q_0(x_K)$.

## B    ALTERNATIVE APPROACH TO MI ESTIMATION

As alluded to earlier in Section 3.1, we can construct an alternative consistent objective function for estimating the pointwise dependence $\frac{p(x,y)}{p(x)p(y)}$. That is, applying the InfoNCE-anchor framework for $q_1(x) \leftarrow p(x,y)$ and $q_0(x, y) \leftarrow p(x)p(y)$, we obtain

$$\mathcal{L}_{K;\nu}^{(2)}(\theta) \triangleq -\frac{K}{K+\nu} \mathbb{E}_{p(x_1,y_1)p(x_2)p(y_2)\cdots p(x_K)p(y_K)} \left[ \log \frac{r_\theta(x_1, y_1)}{\nu + \sum_{i=1}^K r_\theta(x_i, y_i)} \right]$$
$$- \frac{\nu}{K+\nu} \mathbb{E}_{p(x_1)p(y_1)p(x_2)p(y_2)\cdots p(x_K)p(y_K)} \left[ \log \frac{\nu}{\nu + \sum_{i=1}^K r_\theta(x_i, y_i)} \right].$$

While this version results in a different, yet consistent objective function, it is not preferable over the discussed approach in practice.

With this approach, when $\nu = 0$, minimizing $\mathcal{L}_{K;0}^{(2)}(\theta)$ guarantees that for some $C > 0$,

$$r_{\theta^*}(x, y) = C \frac{p(x, y)}{p(x)p(y)}.$$

This is a guarantee analogous to the MLInfoNCE (Song & Ermon, 2020b).

## C    PSEUDOCODE FOR INFONCE-ANCHOR

Here, we provide a pseudocode for the PyTorch implementation of the InfoNCE-anchor objective function.

```python
def infonce_with_anchor(scores, nu=1.0):
    """
    scores: [B, B] tensor where scores[i, j] = f(x_i, y_j)
    nu: prior smoothing hyperparameter
    """
    assert nu > 0.
    device = scores.device
    B = scores.size(0)
    K = B - 1

    # joint term
    mask = torch.zeros(B, B, device=device)
    i = torch.arange(1, B + 1)
    mask[i - 1, i - 2] = -torch.inf
    scores_aug = torch.cat([
        np.log(nu) * torch.ones(B, 1, device=device),
        mask + scores], dim=1)  # augmented score
    joint_term = - (scores.diag().mean() - scores_aug.logsumexp(dim=1).mean())

    # independent term
    neg_inf_diag_mask = torch.zeros(B, B, device=device).fill_diagonal_(-torch.inf)
    scores_aug_neg = torch.cat([
        np.log(nu) * torch.ones(B, 1, device=device),
        neg_inf_diag_mask + scores
        ], dim=1)  # negative augmented score
    marginal_term = - (np.log(nu) - scores_aug_neg.logsumexp(dim=1).mean())

    return (K / (K + nu)) * joint_term + (nu / (K + nu)) * marginal_term
```

We note that, compared to InfoNCE, the additional term in lines 20-26 introduces an additional logsumexp over a $B \times (B + 1)$ tensor, so the loss computation has a slightly larger constant factor compared to standard InfoNCE. However, the dominant cost in our experiments comes from the encoder forward and backward passes and from computing the pairwise score matrix. In this context, the extra logsumexp and concatenation are negligible. Our wall-clock measurements confirm that the end-to-end training time with InfoNCE-anchor is essentially the same as InfoNCE across all settings.

## D    ASYMPTOTIC BEHAVIOR OF INFONCE-ANCHOR

**Theorem 10.** *If $\nu/K \to \beta$ as $K \to \infty$ for some $\beta \geq 0$, then*

$$\lim_{K \to \infty} \left( -\mathcal{L}_{K;\nu}(\theta) + \frac{K \log K}{K + \nu} \right) = \frac{\beta}{\beta + 1} \log \beta + \frac{1}{\beta + 1} \mathbb{E}_{q_1(x)}[\log r_\theta(x)] - \log(\beta + \mathbb{E}_{q_0(x)}[r_\theta(x)])$$

$$\leq \frac{\beta}{\beta + 1} \log \beta + \frac{1}{\beta + 1} D(q_1 \| q_0) - \log(\beta + 1).$$

*The equality holds if and only if $r_\theta(x) = \frac{q_1(x)}{q_0(x)}$ when $\beta > 0$, and $r_\theta(x) = C \frac{q_1(x)}{q_0(x)}$ for some $C > 0$ when $\beta = 0$.*

Rewriting as a lower bound on the KL divergence, we have Eq. (5).

# E   DECISION-THEORETIC TREATMENT OF PROPER SCORING RULES

This section serves as a decision-theoretic foundation on proper scoring rules for conditional probability estimation, which is essential to proving the main statements in Section 3.4, i.e., Proposition 5 and Theorem 6. Appendix E.3 is marked with asterisk, which is included for completeness and can be safely skipped in the first reading.

## E.1   PRELIMINARIES AND DEFINITIONS

We first note that

$$\mathbb{E}_{p(a,z)}[\lambda_z(\hat{\boldsymbol{\eta}}(a))] = \mathbb{E}_{p(a)}\left[\sum_{z=0}^{M}\boldsymbol{\eta}(a)\lambda_z(\hat{\boldsymbol{\eta}}(a))\right] = \mathbb{E}_{p(a)}[\langle\boldsymbol{\eta}(a), \boldsymbol{\lambda}(\hat{\boldsymbol{\eta}}(a))\rangle],$$

which implies that we only need to study the *conditional* problem for each $a$, without the expectation over $a \sim p(a)$. We define the *(conditional) risk* of $\hat{\eta} \in \Delta(\mathcal{Z})$ with respect to $\boldsymbol{\eta}^* \in \Delta(\mathcal{Z})$ as

$$d^{\boldsymbol{\lambda}}(\hat{\boldsymbol{\eta}} \parallel \boldsymbol{\eta}^*) \triangleq \sum_{z \in \mathcal{Z}}(\boldsymbol{\eta}^*)_z \lambda_z(\hat{\boldsymbol{\eta}}) = \langle\boldsymbol{\eta}^*, \boldsymbol{\lambda}(\hat{\boldsymbol{\eta}})\rangle.$$

In particular, we denote

$$f^{\boldsymbol{\lambda}}(\boldsymbol{\eta}^*) \triangleq d^{\boldsymbol{\lambda}}(\boldsymbol{\eta}^* \parallel \boldsymbol{\eta}^*)$$

and call the *pointwise Bayes risk* with respect to $\boldsymbol{\eta}^*$, since we can write the Bayes-optimal risk as

$$\mathbb{E}_{p(a,z)}[f^{\boldsymbol{\lambda}}(\boldsymbol{\eta}(a))] = \mathbb{E}_{p(a,z)}[d^{\boldsymbol{\lambda}}(\boldsymbol{\eta}(a) \parallel \boldsymbol{\eta}(a))] = \min_{\hat{\boldsymbol{\eta}}:\mathcal{A}\to\Delta(\mathcal{Z})}\mathbb{E}_{p(a,z)}[\lambda_z(\hat{\boldsymbol{\eta}}(a))],$$

given that $\boldsymbol{\lambda}$ is proper.

Since the propriety of a scoring rule is independent of the distribution $p(a)$ over $\mathcal{A}$ as alluded to earlier, we can restate the definition of propriety as follows. We define and denote the *regret* of $\hat{\boldsymbol{\eta}}$ with respect to $\boldsymbol{\eta}^*$ under $\boldsymbol{\lambda}$ as

$$\text{Reg}^{\boldsymbol{\lambda}}(\hat{\boldsymbol{\eta}} \parallel \boldsymbol{\eta}^*) \triangleq d^{\boldsymbol{\lambda}}(\hat{\boldsymbol{\eta}} \parallel \boldsymbol{\eta}^*) - d^{\boldsymbol{\lambda}}(\boldsymbol{\eta}^* \parallel \boldsymbol{\eta}^*).$$

**Definition 11.** A loss-function tuple $\boldsymbol{\lambda}$ is said to be *proper* if $\text{Reg}^{\boldsymbol{\lambda}}(\hat{\boldsymbol{\eta}} \parallel \boldsymbol{\eta}^*) \geq 0$ for any $\hat{\boldsymbol{\eta}}, \boldsymbol{\eta}^*$ and $\text{Reg}^{\boldsymbol{\lambda}}(\boldsymbol{\eta}^* \parallel \boldsymbol{\eta}^*) = 0$ for any $\boldsymbol{\eta}^*$. A loss-function tuple $\boldsymbol{\lambda}$ is said to be *strictly proper* if it is proper and $\text{Reg}^{\boldsymbol{\lambda}}(\hat{\boldsymbol{\eta}} \parallel \boldsymbol{\eta}^*) = 0$ if and only if $\hat{\boldsymbol{\eta}} = \boldsymbol{\eta}^*$, for any $\boldsymbol{\eta}^*$.

We now state the characterization of differentiable (strictly) proper loss functions. If $\boldsymbol{\lambda}$ is differentiable, we let

$$\mathbf{g}^{\boldsymbol{\lambda}}(\boldsymbol{\eta}^*) \triangleq \nabla_{\hat{\boldsymbol{\eta}}}d^{\boldsymbol{\lambda}}(\hat{\boldsymbol{\eta}} \parallel \boldsymbol{\eta}^*)|_{\hat{\boldsymbol{\eta}}=\boldsymbol{\eta}^*} = (\langle\nabla_j\boldsymbol{\lambda}(\boldsymbol{\eta}^*), \boldsymbol{\eta}^*\rangle)_{j\in\mathcal{Z}} = \mathbf{J}\boldsymbol{\lambda}(\boldsymbol{\eta}^*)\boldsymbol{\eta}^*, \tag{11}$$

which is the gradient of the pointwise risk function $\hat{\boldsymbol{\eta}} \mapsto d^{\boldsymbol{\lambda}}(\hat{\boldsymbol{\eta}} \parallel \boldsymbol{\eta}^*)$ at $\hat{\boldsymbol{\eta}} = \boldsymbol{\eta}^*$. Here, $\mathbf{J}\boldsymbol{\lambda}(\boldsymbol{\eta}^*) \in \mathbb{R}^{m\times m}$ denotes the Jacobian of the matrix $\boldsymbol{\lambda}: \Delta(\mathcal{Z}) \to \mathbb{R}^m$, i.e.,

$$(\mathbf{J}\boldsymbol{\lambda}(\boldsymbol{\eta}^*))_{ij} \triangleq \frac{\partial\lambda_j(\boldsymbol{\eta}^*)}{\partial\eta_i^*}.$$

**Theorem 12.** *A scoring rule $\boldsymbol{\lambda} = (\lambda_z: \Delta(\mathcal{Z}) \to \mathbb{R})_{z\in\mathcal{Z}}$ is (strictly) proper if and only if (1) the pointwise Bayes risk function $\boldsymbol{\eta} \mapsto f^{\boldsymbol{\lambda}}(\boldsymbol{\eta})$ is (strictly) concave over $\Delta(\mathcal{Z})$ and (2) $\mathbf{g}^{\boldsymbol{\lambda}}(\boldsymbol{\eta}^*) = \mathbf{0}$ for any $\boldsymbol{\eta}^* \in \Delta(\mathcal{Z})$.*

To prove the theorem, we first state a key technical lemma. Given a differentiable function $f: V \to \mathbb{R}$ over a subset $V$ of an Euclidean space, we define the *Bregman distortion* $B_f: V \times V \to \mathbb{R}$ as

$$B_f(u, v) \triangleq f(u) - f(v) - \langle\nabla f(v), u - v\rangle.$$

If $f$ is convex, $B_f(x, z)$ is called the *Bregman divergence* generated by $f$. Finally, let $\overline{f}^{\boldsymbol{\lambda}}(\boldsymbol{\eta}) \triangleq -f^{\boldsymbol{\lambda}}(\boldsymbol{\eta})$ denote the negative pointwise Bayes risk.

**Lemma 13.** *For any $\hat{\boldsymbol{\eta}}, \boldsymbol{\eta}^*$, we have*

$$\mathsf{Reg}^{\boldsymbol{\lambda}}(\hat{\boldsymbol{\eta}} \parallel \boldsymbol{\eta}^*) = B_{\overline{f}^{\boldsymbol{\lambda}}}(\boldsymbol{\eta}^*, \hat{\boldsymbol{\eta}}) + \langle \mathbf{g}^{\boldsymbol{\lambda}}(\hat{\boldsymbol{\eta}}), \hat{\boldsymbol{\eta}} - \boldsymbol{\eta}^* \rangle.$$

*Proof.* By chain rule, we have

$$\nabla f^{\boldsymbol{\lambda}}(\hat{\boldsymbol{\eta}}) = \boldsymbol{\lambda}(\hat{\boldsymbol{\eta}}) + \mathbf{g}^{\boldsymbol{\lambda}}(\hat{\boldsymbol{\eta}}).$$

Therefore, by the definition of Bregman distortion, we have

$$
\begin{aligned}
B_{\overline{f}^{\boldsymbol{\lambda}}}(\boldsymbol{\eta}^*, \hat{\boldsymbol{\eta}}) &= -B_{f^{\boldsymbol{\lambda}}}(\boldsymbol{\eta}^*, \hat{\boldsymbol{\eta}}) \\
&= -f^{\boldsymbol{\lambda}}(\boldsymbol{\eta}^*) + f^{\boldsymbol{\lambda}}(\hat{\boldsymbol{\eta}}) + \langle \nabla f^{\boldsymbol{\lambda}}(\hat{\boldsymbol{\eta}}), \boldsymbol{\eta}^* - \hat{\boldsymbol{\eta}} \rangle \\
&= -d^{\boldsymbol{\lambda}}(\boldsymbol{\eta}^* \parallel \boldsymbol{\eta}^*) + d^{\boldsymbol{\lambda}}(\hat{\boldsymbol{\eta}} \parallel \hat{\boldsymbol{\eta}}) - \langle \boldsymbol{\lambda}(\hat{\boldsymbol{\eta}}) + \mathbf{g}^{\boldsymbol{\lambda}}(\hat{\boldsymbol{\eta}}), \boldsymbol{\eta}^* - \hat{\boldsymbol{\eta}} \rangle \\
&= -d^{\boldsymbol{\lambda}}(\boldsymbol{\eta}^* \parallel \boldsymbol{\eta}^*) + d^{\boldsymbol{\lambda}}(\hat{\boldsymbol{\eta}} \parallel \hat{\boldsymbol{\eta}}) + d^{\boldsymbol{\lambda}}(\hat{\boldsymbol{\eta}} \parallel \boldsymbol{\eta}^*) - d^{\boldsymbol{\lambda}}(\hat{\boldsymbol{\eta}} \parallel \hat{\boldsymbol{\eta}}) + \langle \mathbf{g}^{\boldsymbol{\lambda}}(\hat{\boldsymbol{\eta}}), \boldsymbol{\eta}^* - \hat{\boldsymbol{\eta}} \rangle \\
&= \mathsf{Reg}^{\boldsymbol{\lambda}}(\hat{\boldsymbol{\eta}} \parallel \boldsymbol{\eta}^*) + \langle \mathbf{g}^{\boldsymbol{\lambda}}(\hat{\boldsymbol{\eta}}), \boldsymbol{\eta}^* - \hat{\boldsymbol{\eta}} \rangle,
\end{aligned}
$$

which concludes the proof. $\square$

The first condition (1) imposes that the estimation problem becomes (strictly) not easier as we *mix* the class probabilities. The second condition (2) formalizes that if $\boldsymbol{\eta}^*$ is the underlying class probability, then $\boldsymbol{\eta} = \boldsymbol{\eta}^*$ is a local minimizer of the conditional risk function $\boldsymbol{\eta} \mapsto d^{\boldsymbol{\lambda}}(\boldsymbol{\eta} \parallel \boldsymbol{\eta}^*)$.

Now we are ready to prove Theorem 12.

*Proof of Theorem 12.* We first prove the only-if direction. If $\boldsymbol{\lambda}$ is proper, then $d^{\boldsymbol{\lambda}}(\hat{\boldsymbol{\eta}} \parallel \boldsymbol{\eta}^*) \geq d^{\boldsymbol{\lambda}}(\boldsymbol{\eta}^* \parallel \boldsymbol{\eta}^*)$ for any $\hat{\boldsymbol{\eta}}$ and $\boldsymbol{\eta}^*$ by definition. That is, $\hat{\boldsymbol{\eta}} \mapsto d^{\boldsymbol{\lambda}}(\hat{\boldsymbol{\eta}} \parallel \boldsymbol{\eta}^*)$ is stationary at $\hat{\boldsymbol{\eta}} = \boldsymbol{\eta}^*$, and thus the gradient $\mathbf{g}^{\boldsymbol{\lambda}}(\boldsymbol{\eta}) = \nabla_{\hat{\boldsymbol{\eta}}} d^{\boldsymbol{\lambda}}(\hat{\boldsymbol{\eta}} \parallel \boldsymbol{\eta})|_{\hat{\boldsymbol{\eta}}=\boldsymbol{\eta}} = 0$ for any $\boldsymbol{\eta}$. Now, by the identity in Lemma 13, we have $B_{\overline{f}^{\boldsymbol{\lambda}}}(\boldsymbol{\eta}^*, \hat{\boldsymbol{\eta}}) = \mathsf{Reg}^{\boldsymbol{\lambda}}(\hat{\boldsymbol{\eta}} \parallel \boldsymbol{\eta}^*) \geq 0$ for any $\boldsymbol{\eta}^*, \hat{\boldsymbol{\eta}}$, which implies that the function $f^{\boldsymbol{\lambda}} = -\overline{f}^{\boldsymbol{\lambda}}$ is concave. Further, if $\boldsymbol{\lambda}$ is strictly proper, then $B_{\overline{f}^{\boldsymbol{\lambda}}}(\boldsymbol{\eta}^*, \hat{\boldsymbol{\eta}}) = \mathsf{Reg}^{\boldsymbol{\lambda}}(\hat{\boldsymbol{\eta}} \parallel \boldsymbol{\eta}^*) > 0$ for any $\hat{\boldsymbol{\eta}} \neq \boldsymbol{\eta}^*$, which implies that $f^{\boldsymbol{\lambda}}$ is strictly concave.

For the converse, i.e., the if direction, we can directly apply the identity in Lemma 13 and conclude $\mathsf{Reg}^{\boldsymbol{\lambda}}(\hat{\boldsymbol{\eta}} \parallel \boldsymbol{\eta}^*) \geq B_{\overline{f}^{\boldsymbol{\lambda}}}(\boldsymbol{\eta}^*, \hat{\boldsymbol{\eta}}) \geq 0$ by the convexity of $\overline{f}^{\boldsymbol{\lambda}}$. It is clear that $\boldsymbol{\lambda}$ is strictly proper if $\overline{f}^{\boldsymbol{\lambda}}$ is strictly concave. $\square$

### E.2 FROM LOSS FUNCTION TO GENERATING FUNCTION

Given a loss function $\boldsymbol{\lambda}$, we define a corresponding *generating function*

$$\Psi^{\boldsymbol{\lambda}}(\boldsymbol{\rho}) \triangleq -\langle \boldsymbol{\rho}, \boldsymbol{\lambda}(\boldsymbol{\eta}) \rangle$$

for $\boldsymbol{\rho} \in \{1\} \times \mathbb{R}_+^M$, so that we can write the pointwise Bayes risk at $\boldsymbol{\eta}^*$ as

$$f^{\boldsymbol{\lambda}}(\boldsymbol{\eta}^*) = d^{\boldsymbol{\lambda}}(\boldsymbol{\eta}^* \parallel \boldsymbol{\eta}^*) = -\eta_0^* \Psi^{\boldsymbol{\lambda}}(\boldsymbol{\rho}^*).$$

Then, it is easy to check that

**Proposition 14.** *If $\boldsymbol{\lambda}$ is (strictly) proper, $\boldsymbol{\rho} \mapsto \Psi^{\boldsymbol{\lambda}}(\boldsymbol{\rho})$ is (strictly) convex.*

*Proof.* If $\boldsymbol{\lambda}$ is (strictly) proper, then the negative pointwise Bayes risk function $\boldsymbol{\eta}^* \mapsto -f^{\boldsymbol{\lambda}}(\boldsymbol{\eta}^*) = \eta_0^* \Psi^{\boldsymbol{\lambda}}(\boldsymbol{\rho}^*)$ is (strictly) convex by Theorem 12. Since the mapping is a perspective function of $\boldsymbol{\rho} \mapsto \Psi^{\boldsymbol{\lambda}}(\boldsymbol{\rho})$, $\Psi^{\boldsymbol{\lambda}}$ must be (strictly) convex. $\square$

**Remark 15** (From generating function to loss function). Conversely, we can define a loss function from a differentiable function $\Psi \colon \{1\} \times \mathbb{R}_+^M$ as follows:

$$\boldsymbol{\lambda}^{\Psi}(\boldsymbol{\eta}) \triangleq \begin{bmatrix} \langle \nabla \Psi(\boldsymbol{\rho}), \boldsymbol{\rho} \rangle - \Psi(\boldsymbol{\rho}) \\ -\nabla \Psi(\boldsymbol{\rho})_{1:M} \end{bmatrix},$$

so that we can write the pointwise Bayes risk at $\boldsymbol{\eta}^*$ as

$$f^{\boldsymbol{\lambda}^{\Psi}}(\boldsymbol{\eta}^*) = d^{\boldsymbol{\lambda}^{\Psi}}(\boldsymbol{\eta}^* \parallel \boldsymbol{\eta}^*) = -\eta_0^* \Psi(\boldsymbol{\rho}^*).$$

### E.3 ONE-TO-ONE CORRESPONDENCE*

A natural question to ask is whether $\boldsymbol{\lambda} \mapsto \Psi^{\boldsymbol{\lambda}}$ and $\Psi \mapsto \boldsymbol{\lambda}^{\Psi}$ are inverse mappings each other. Indeed, we have the following propositions.

**Proposition 16.**

$$\boldsymbol{\lambda}^{\Psi^{\boldsymbol{\lambda}}}(\boldsymbol{\eta}) = \boldsymbol{\lambda}(\boldsymbol{\eta}) - \langle \boldsymbol{\eta}, \mathbf{g}^{\boldsymbol{\lambda}}(\boldsymbol{\eta}) \rangle \mathbb{1} + \mathbf{g}^{\boldsymbol{\lambda}}(\boldsymbol{\eta}).$$

Hence, in particular, if $\boldsymbol{\lambda}$ is proper, it readily follows that $\boldsymbol{\lambda}^{\Psi^{\boldsymbol{\lambda}}}(\boldsymbol{\eta}) \equiv \boldsymbol{\lambda}(\boldsymbol{\eta})$.

*Proof of Proposition 16.* First, we consider $z \in \{1, \dots, M\}$. Note that

$$\frac{\partial \lambda_z(\boldsymbol{\eta})}{\partial \rho_z} = \Big\langle \frac{\partial}{\partial \rho_z} \frac{(1, \rho_1, \dots, \rho_M)}{1 + \rho_1 + \dots + \rho_M}, \nabla \lambda_z(\boldsymbol{\eta}) \Big\rangle$$
$$= \eta_0 \langle -\boldsymbol{\eta} + \mathbf{e}_Y, \nabla \lambda_z(\boldsymbol{\eta}) \rangle,$$

for any $z = 0, 1, \dots, M$. Thus, we have

$$\sum_{z=0}^{M} \rho_z \frac{\partial \lambda_z(\boldsymbol{\eta})}{\partial \rho_z} = \eta_0 \sum_z \rho_z \langle -\boldsymbol{\eta} + \mathbf{e}_Y, \nabla \lambda_z(\boldsymbol{\eta}) \rangle$$
$$= \Big\langle -\boldsymbol{\eta} + \mathbf{e}_Y, \sum_z \eta_z \nabla \lambda_z(\boldsymbol{\eta}) \Big\rangle$$
$$= \langle -\boldsymbol{\eta} + \mathbf{e}_Y, \mathbf{g}^{\boldsymbol{\lambda}}(\boldsymbol{\eta}) \rangle.$$

This implies that

$$\lambda_z^{\Psi^{\boldsymbol{\lambda}}}(\boldsymbol{\eta}) = \frac{\partial \Psi^{\boldsymbol{\lambda}}(\boldsymbol{\eta})}{\partial \rho_z}$$
$$= -\frac{\partial \lambda_0(\boldsymbol{\eta})}{\partial \rho_z} - \lambda_z(\boldsymbol{\eta}) - \sum_{z=1}^{M} \rho_z \frac{\partial \lambda_z(\boldsymbol{\eta})}{\partial \rho_z}$$
$$= -\lambda_z(\boldsymbol{\eta}) - \sum_{z=0}^{M} \rho_z \frac{\partial \lambda_z(\boldsymbol{\eta})}{\partial \rho_z}$$
$$= -\lambda_z(\boldsymbol{\eta}) - \langle \boldsymbol{\eta} + \mathbf{e}_z, \mathbf{g}^{\boldsymbol{\lambda}}(\boldsymbol{\eta}) \rangle.$$

We now consider $z = 0$. Observe that

$$\langle \nabla \Psi^{\boldsymbol{\lambda}}(\boldsymbol{\rho}), \boldsymbol{\rho} \rangle = \sum_{z=1}^{M} \rho_z \frac{\partial \Psi^{\boldsymbol{\lambda}}(\boldsymbol{\rho})}{\partial \rho_z}$$
$$= -\sum_{z=1}^{M} \rho_z \lambda_z(\boldsymbol{\eta}) - \langle \boldsymbol{\eta} + \mathbf{e}_0, \mathbf{g}^{\boldsymbol{\lambda}}(\boldsymbol{\eta}) \rangle.$$

Hence, we have

$$\lambda_0^{\Psi^{\boldsymbol{\lambda}}}(\boldsymbol{\eta}) = \langle \nabla \Psi^{\boldsymbol{\lambda}}(\boldsymbol{\rho}), \boldsymbol{\rho} \rangle - \Psi^{\boldsymbol{\lambda}}(\boldsymbol{\rho})$$
$$= \lambda_0(\boldsymbol{\eta}) + \sum_{z=1}^{M} \rho_z \lambda_z(\boldsymbol{\eta}) - \sum_{z=1}^{M} \rho_z \lambda_z(\boldsymbol{\eta}) - \langle \boldsymbol{\eta} + \mathbf{e}_0, \mathbf{g}^{\boldsymbol{\lambda}}(\boldsymbol{\eta}) \rangle$$
$$= \lambda_0(\boldsymbol{\eta}) - \langle \boldsymbol{\eta} + \mathbf{e}_0, \mathbf{g}^{\boldsymbol{\lambda}}(\boldsymbol{\eta}) \rangle.$$

This concludes the proof. $\qquad \square$

The following statement asserts that the generating function induced by the induced loss function of a generating function corresponds to the original generating function.

**Proposition 17.**

$$\Psi^{\boldsymbol{\lambda}^{\Psi}}(\boldsymbol{\rho}) \equiv \Psi(\boldsymbol{\rho}).$$

*Proof.* By definition, it is easy to check that

$$\Psi^{\boldsymbol{\lambda}^{\Psi}}(\boldsymbol{\rho}) = -\langle \boldsymbol{\rho}, \boldsymbol{\lambda}^{\Psi}(\boldsymbol{\rho})\rangle$$

$$= -(\langle \nabla \Psi(\boldsymbol{\rho}), \boldsymbol{\rho}\rangle - \Psi(\boldsymbol{\rho})) + \sum_{z=1}^{M} \rho_z \frac{\partial \Psi(\boldsymbol{\rho})}{\partial \rho_z}$$

$$= \Psi(\boldsymbol{\rho}). \qquad \square$$

Therefore, there is a one-to-one correspondence between (strictly) proper loss functions $\{\boldsymbol{\lambda} \colon \Delta(\mathcal{Z}) \to \mathbb{R}^{\mathcal{Z}}\}$ and (strictly) convex functions $\{\Psi \colon \{1\} \times \mathbb{R}_+^M \to \mathbb{R}\}$.

### E.4 CONNECTION TO BREGMAN DIVERGENCES

Note the following proposition.

**Proposition 18.**

$$B_{f^{\boldsymbol{\lambda}}}(\boldsymbol{\eta}^*, \boldsymbol{\eta}) = -\eta_0^* B_{\Psi^{\boldsymbol{\lambda}}}(\boldsymbol{\rho}^*, \boldsymbol{\rho}).$$

The following corollary reveals that any CPE objective function induced by a proper scoring rule can be understood as a Bregman divergence minimization.

**Corollary 19.** *If $\boldsymbol{\lambda}$ is proper, then*

$$\mathsf{Reg}^{\boldsymbol{\lambda}}(\boldsymbol{\eta} \parallel \boldsymbol{\eta}^*) = \eta_0^* B_{\Psi^{\boldsymbol{\lambda}}}(\boldsymbol{\rho}^*, \boldsymbol{\rho}).$$

In other words, it shows that a proper loss function $\boldsymbol{\lambda}$ acts only through the form of the Bregman divergence $B_{\Psi^{\boldsymbol{\lambda}}}(\cdot, \cdot)$. In other words, $\boldsymbol{\lambda}$ and $\boldsymbol{\lambda}'$ are equivalent CPE loss functions if $B_{\Psi^{\boldsymbol{\lambda}}}(\cdot, \cdot) \equiv B_{\Psi^{\boldsymbol{\lambda}'}}(\cdot, \cdot)$. This defines an equivalence class in the set of loss functions

$$\Lambda(\Psi) \triangleq \{\boldsymbol{\lambda} \colon \Delta(\mathcal{Z}) \to \mathbb{R}^{\mathcal{Z}} | B_{\Psi^{\boldsymbol{\lambda}}}(\cdot, \cdot) \equiv B_{\Psi}(\cdot, \cdot)\}.$$

We know that this set is always not empty, since Proposition 17 implies that

$$\boldsymbol{\lambda}^{\Psi} \in \Lambda(\Psi).$$

Consider a subset

$$\Lambda_o(\Psi) \triangleq \{\boldsymbol{\lambda} \colon \Delta(\mathcal{Z}) \to \mathbb{R}^{\mathcal{Z}} | \boldsymbol{\lambda} \in \Lambda(\Psi), \mathbf{g}^{\boldsymbol{\lambda}}(\boldsymbol{\eta}) \equiv \mathbf{0}\}.$$

The loss functions in this subset can be thought as *canonical* functions, as we require $\mathbf{g}^{\boldsymbol{\lambda}}(\boldsymbol{\eta}) \equiv \mathbf{0}$ to check propriety in Theorem 12. Note that

$$\boldsymbol{\lambda}^{\Psi} \in \Lambda_o(\Psi),$$

since Lemma 24 establishes that $\mathbf{g}^{\boldsymbol{\lambda}^{\Psi}}(\boldsymbol{\eta}) \equiv \mathbf{0}$. A small open question is whether $\boldsymbol{\lambda}^{\Psi}$ is an unique element of $\Lambda_o(\Psi)$.

**Remark 20** (Properization). We remark that for any $\boldsymbol{\lambda} \in \Lambda(\Psi)$, we can map it to another element $\boldsymbol{\lambda}' \in \Lambda_o(\Psi)$, by defining it as

$$\boldsymbol{\lambda}'(\boldsymbol{\eta}) \triangleq \boldsymbol{\lambda}(\boldsymbol{\eta}) + \mathbf{g}^{\boldsymbol{\lambda}}(\boldsymbol{\eta}) - \langle \boldsymbol{\eta}, \mathbf{g}^{\boldsymbol{\lambda}}(\boldsymbol{\eta})\rangle \mathbb{1}.$$

It is easy to check that $\boldsymbol{\lambda}' \in \Lambda_o(\Psi)$ indeed. One can think of this as a *properization* of a loss function $\boldsymbol{\lambda}$, since for a convex function $\Psi$, any loss function $\boldsymbol{\lambda} \in \Lambda(\Psi)$ can be made into a proper loss $\boldsymbol{\lambda}' \in \Lambda_o(\Psi)$.

### E.5 EXAMPLES OF PROPER SCORING RULES

We first start with proper *binary* scoring rules; see Table 3. Most of the examples can be found from (Gneiting & Raftery, 2007). We refer to rules generated from the $\Psi$-induced scoring rules (Eq. (8)) by *asymmetric* scoring rules, and the $\Phi$-induced rules (Eq. (12)) by *symmetric* rules.

---

[2]If $\alpha = 2$, famously known as the Brier score (Brier, 1950; Gneiting & Raftery, 2007).

[3]Called the spherical score when $\alpha = 2$ (Gneiting & Raftery, 2007). When $\alpha \to 1$, boils down to the log score.

Table 3: Examples of strictly proper binary scoring rules.

| Asymmetric scoring rule | $\Psi(\rho)$ | $\lambda_0^\Psi(\boldsymbol{\eta}), \lambda_1^\Psi(\boldsymbol{\eta})$ (see Eq. (8)) |
|---|---|---|
| KLIEP (Sugiyama et al., 2008) | $\rho \log \rho$ | $\frac{1}{\eta_0}, -\log\frac{\eta_1}{\eta_0}$ |
| Robust DRE ($\alpha \notin \{0,1\}$) (Sugiyama et al., 2012) | $\frac{\rho^\alpha}{\alpha(\alpha-1)}$ or $\frac{\rho^\alpha - \rho}{\alpha(\alpha-1)}$ | $\frac{1}{\alpha}\frac{\eta_0^\alpha + \eta_1^\alpha}{\eta_0^\alpha} + \frac{1}{\alpha(\alpha-1)}, \frac{1}{1-\alpha}(\frac{\eta_1}{\eta_0})^{\alpha-1}$ |
| Inverse log | $-\log\rho$ | $\log\frac{\eta_1}{\eta_0} - 1, \frac{\eta_0}{\eta_1}$ |
| Symmetric scoring rule | $\Phi(\eta_0, \eta_1)$ | $\lambda_z^{\Psi_\Phi}(\boldsymbol{\eta})$ (see Eq. (12)) |
| Log (Good, 1952) | $\eta_0 \log\eta_0 + \eta_1 \log\eta_1$ | $-\log\eta_z$ |
| Power ($\alpha \notin \{0,1\}$)[2] | $\frac{\eta_0^\alpha + \eta_1^\alpha - 1}{\alpha(\alpha-1)}$ | $\frac{\eta_0^\alpha + \eta_1^\alpha}{\alpha} - \frac{\eta_z^{\alpha-1}}{\alpha-1}$ |
| Sym. inverse log | $-\log\eta_0 - \log\eta_1$ | $\log\eta_0 + \log\eta_1 + \frac{1}{\eta_z}$ |
| Pseudo-spherical ($\alpha \notin \{0,1\}$) (Gneiting & Raftery, 2007)[3] | $\frac{1}{\alpha-1}(\frac{\eta_0^\alpha + \eta_1^\alpha}{2})^{\frac{1}{\alpha}}$ | $-\frac{2^{-\frac{1}{\alpha}}}{\alpha-1}(\frac{\eta_z}{(\eta_0^\alpha+\eta_1^\alpha)^{\frac{1}{\alpha}}})^{\alpha-1}$ |

Now, by naturally extending the definition of the elementary generating functions for the binary scoring rules, we can derive their multi-ary counterparts as shown in Table 4. We note that the multi-ary asymmetric scoring rules, when considered with our binary density ratio estimation framework below, boil down to the ones induced by the binary scoring rules. Therefore, since the nontrivial examples are from extending the symmetric scoring rules, we omit the multiary version of asymmetric rules.

Table 4: Examples of *symmetric* strictly proper $(M+1)$-ary scores.

| Symmetric scoring rule | $\Phi(\boldsymbol{\eta})$ | $\lambda_z^{\Psi_\Phi}(\boldsymbol{\eta})$ (see Eq. (12)) | Known as |
|---|---|---|---|
| Log | $\langle\boldsymbol{\eta}, \log\boldsymbol{\eta}\rangle$ | $-\log\eta_z$ | |
| Power ($\alpha \notin \{0,1\}$) | $\frac{\|\boldsymbol{\eta}\|_\alpha^\alpha}{\alpha(\alpha-1)}$ or $\frac{\|\boldsymbol{\eta}\|_\alpha^{\alpha-1}}{\alpha(\alpha-1)}$ | $\frac{\|\boldsymbol{\eta}\|_\alpha^\alpha}{\alpha} - \frac{\eta_z^{\alpha-1}}{\alpha-1}$ | Tsallis scoring rule (Dawid & Musio, 2014) |
| Sym. inverse log | $-\sum_{z=0}^M \log\eta_z$ | $\sum_{z=0}^M \log\eta_z + \frac{1}{\eta_z}$ | |
| Spherical ($\alpha \notin \{0,1\}$) | $\frac{(M+1)^{-\frac{1}{\alpha}}}{\alpha-1}\|\boldsymbol{\eta}\|_\alpha$ | $-\frac{(M+1)^{-\frac{1}{\alpha}}}{\alpha-1}(\frac{\eta_z}{\|\boldsymbol{\eta}\|_\alpha})^{\alpha-1}$ | |

# F  DETAILS ON EXTENSIONS WITH PROPER SCORING RULES

In this section, we provide technical materials deferred from Section 3.4 on the extensions with proper scoring rules.

## F.1  ALTERNATIVE CHARACTERIZATION OF PROPER SCORING RULE

An alternative, yet equivalent representation of a proper scoring rule is based on a convex function $\Phi(\boldsymbol{\eta})$ over $\boldsymbol{\eta} \in \Delta([0:M])$. One can induce a convex function $\Psi(\boldsymbol{\rho})$ from a convex function $\Phi(\boldsymbol{\eta})$ by the perspective transformation:

$$\Psi_\Phi(\boldsymbol{\rho}) \triangleq (1 + \rho_1 + \ldots + \rho_M)\Phi\left(\frac{[1; \boldsymbol{\rho}]}{1 + \rho_1 + \ldots + \rho_M}\right).$$

**Theorem 21.** *Given a differentiable function* $\Phi \colon \Delta([0:M]) \to \mathbb{R}$,

$$\boldsymbol{\lambda}^{\Psi_\Phi}(\boldsymbol{\eta}) = \Big(\langle\boldsymbol{\eta}, \nabla_{\boldsymbol{\eta}}\Phi(\boldsymbol{\eta})\rangle - \Phi(\boldsymbol{\eta})\Big)\mathbf{1} - \nabla_{\boldsymbol{\eta}}\Phi(\boldsymbol{\eta}). \tag{12}$$

*Proof.* First, we can write

$$\mathcal{L}_{K;\nu}^{\Psi_\Phi}(\boldsymbol{\eta}_\theta) - \mathcal{L}_{K;\nu}^{\Psi_\Phi}(\boldsymbol{\eta}^*) = \mathbb{E}_{p(x_{1:K})}\Big[\langle\boldsymbol{\eta}^*(x_{1:K}), \boldsymbol{\lambda}^{\Psi_\Phi}(\boldsymbol{\eta}_\theta(x_{1:K}))\rangle - \langle\boldsymbol{\eta}^*(x_{1:K}), \boldsymbol{\lambda}^{\Psi_\Phi}(\boldsymbol{\eta}^*(x_{1:K}))\rangle\Big].$$

It is easy to check, from the definition of the $\Psi_\Phi$-induced scoring rule in Eq. (12),

$$\langle \boldsymbol{\eta}^*, \boldsymbol{\lambda}^{\Psi_\Phi}(\boldsymbol{\eta}_\theta) \rangle = -\Phi(\boldsymbol{\eta}_\theta) - \langle \nabla_{\boldsymbol{\eta}} \Phi(\boldsymbol{\eta}_\theta), \boldsymbol{\eta}^* - \boldsymbol{\eta}_\theta \rangle.$$

In particular,

$$\langle \boldsymbol{\eta}^*, \boldsymbol{\lambda}^{\Psi_\Phi}(\boldsymbol{\eta}^*) \rangle = -\Phi(\boldsymbol{\eta}^*).$$

Hence, we have

$$\langle \boldsymbol{\eta}^*, \boldsymbol{\lambda}^{\Psi_\Phi}(\boldsymbol{\eta}_\theta) \rangle - \langle \boldsymbol{\eta}^*, \boldsymbol{\lambda}^{\Psi_\Phi}(\boldsymbol{\eta}^*) \rangle = B_\Phi(\boldsymbol{\eta}^*, \boldsymbol{\eta}_\theta). \qquad \square$$

See the proof of Theorem 6 in Appendix G for a comparison.

We remark that, for a (strictly) convex function $\Phi$, $\Psi$ is (strictly) convex since the perspective transformation preserves the convexity, and thus

$$\langle \boldsymbol{\eta}^*, \boldsymbol{\lambda}^{\Psi_\Phi}(\boldsymbol{\eta}) \rangle \geq \langle \boldsymbol{\eta}^*, \boldsymbol{\lambda}^{\Psi_\Phi}(\boldsymbol{\eta}^*) \rangle = -\Phi(\boldsymbol{\eta}^*).$$

We note that the right hand side is the *Bayes optimal risk*. In other words, a convex function $\Phi(\cdot)$ can characterize a proper scoring rule as its (negative) Bayes-optimal risk.

**Theorem 22.** *For $\nu \geq 0$,*

$$\mathcal{L}_{K;\nu}^{\Psi_\Phi}(\boldsymbol{\eta}_\theta) - \mathcal{L}_{K;\nu}^{\Psi_\Phi}(\boldsymbol{\eta}^*) = \mathbb{E}_{p(x_{1:K})}\Big[ B_\Phi\big(\boldsymbol{\eta}^*(x_{1:K}), \boldsymbol{\eta}_\theta(x_{1:K})\big)\Big],$$

*For $\nu \geq 0$ and a convex function $\Phi$, we have*

$$-\mathcal{L}_{K;\nu}^{\Psi_\Phi}(\boldsymbol{\eta}_\theta) \leq -\mathcal{L}_{K;\nu}^{\Psi_\Phi}(\boldsymbol{\eta}^*) = \mathbb{E}_{p(x_{1:K})}\Big[\Phi\big(\boldsymbol{\eta}^*(x_{1:K})\big)\Big].$$

*If $\nu > 0$, for a strictly convex function $\Phi$, the equality is achieved if and only if $r_\theta(x) = \frac{q_1(x)}{q_0(x)}$.*

If $\nu = 0$, i.e., if there is no anchor class 0, we can only estimate the density ratio up to a multiplicative constant, as the original InfoNCE guarantees.

### F.2 ON IMPLEMENTATION

Here, we We say that a scoring rule $\boldsymbol{\lambda}$ is $\{y_1, y_2\}$-*invariant* for $y_1 \neq y_2 \in \mathcal{Y}$ if $\lambda_{y_1}(\boldsymbol{\eta}) = \lambda_{y_2}(\boldsymbol{\eta}')$ and $\lambda_{y_2}(\boldsymbol{\eta}) = \lambda_{y_1}(\boldsymbol{\eta}')$ for any $\boldsymbol{\eta}, \boldsymbol{\eta}'$ such that $\eta_y = \eta'_y$ for $y \notin \{y_1, y_2\}$ and $\eta_{y_1} = \eta'_{y_2}$ and $\eta_{y_2} = \eta'_{y_1}$.

**Proposition 23.** *If the scoring rule $\boldsymbol{\lambda}$ is $\{z_1, z_2\}$-invariant for any $\{z_1, z_2\} \subseteq \{1, 2, \ldots, K\}$, we have*

$$\mathcal{L}_{K;\nu}^{\Psi}(\boldsymbol{\eta}_\theta) = \frac{K}{K+\nu}\mathbb{E}_{q_1(x_1)q_0(x_2)\cdots q_0(x_K)}[\lambda_1^\Psi(\boldsymbol{\eta}_\theta(x_{1:K}))]$$
$$+ \frac{\nu}{K+\nu}\mathbb{E}_{q_0(x_1)q_0(x_2)\cdots q_0(x_K)}[\lambda_0^\Psi(\boldsymbol{\eta}_\theta(x_{1:K}))]. \tag{13}$$

### F.3 EXAMPLES OF INFONCE-ANCHOR-TYPE DRE OBJECTIVES

Recall the examples of proper scoring rules in Appendix E.5. In Table 5, we first list the canonical consistent DRE objectives derived by asymmetric scoring rules (see Table 3). As noted earlier, the tensorization of InfoNCE-anchor does not have any effect with asymmetric scoring rules, and the objectives boil down to the standard binary DRE objectives.

As noted in the last column of the table, these binary DRE objectives have been extensively used and studied in the various literature on DRE, MI estimation, and representation learning. We mention in passing that a recent paper (Ryu et al., 2025), building on noise-contrastive estimation (Gutmann & Hyvärinen, 2012), revealed a connection between these rules and the maximum likelihood estimation principle.

In Table 6, we list the InfoNCE-anchor-type objectives based on the symmetric scoring rules (see Table 4). Table 7 lists the corresponding InfoNCE-type objectives (i.e., without anchor). We remark that the Spherical objective in the main text corresponds to the last row in Table 6.

Table 5: Examples of consistent DRE objectives derived from *asymmetric* scoring rules (first half of Table 3). Note that these objectives induced by asymmetric scoring rules do not depend on $K$ and $\nu$.

| Asym. scoring rule | $\mathcal{L}_{K;\nu}^{\Psi}(\boldsymbol{\eta}_\theta)$ (see Eq. (13)) | Known as |
|---|---|---|
| Log | $\mathbb{E}_{q_1}[-\log r_\theta(x)] + \mathbb{E}_{q_0}[r_\theta(x)]$ | KLIEP (Sugiyama et al., 2008) in DRE. NWJ (Nguyen et al., 2010) in MI estimation. |
| Power ($\alpha \notin (0,1)$) | $\mathbb{E}_{q_1}[\frac{r_\theta(x)^{\alpha-1}}{1-\alpha}] + \mathbb{E}_{q_0}[\frac{r_\theta(x)^\alpha}{\alpha}]$ | Robust DRE (Sugiyama et al., 2012), KLIEP (Sugiyama et al., 2008) when $\alpha \to 1$, LSIF (Kanamori et al., 2009) when $\alpha = 2$ in DRE. |
| (when $\alpha = 2$) | $-\mathbb{E}_{q_1}[r_\theta(x)] + \frac{1}{2}\mathbb{E}_{q_0}[r_\theta(x)^2]$ | In MI estimation/DRE, known as $\chi^2$ or DRF (Tsai et al., 2020). In rep. learning, H-score (Wang et al., 2019), spectral contrastive loss (HaoChen et al., 2021), CCA (Chapman et al., 2024), LoRA loss (Ryu et al., 2024). |
| Inverse log | $\mathbb{E}_{q_1}[\frac{1}{r_\theta(x)}] + \mathbb{E}_{q_0}[\log r_\theta(x)]$ | |

Table 6: Examples of InfoNCE-anchor-type DRE objectives ($\nu > 0$), derived from *symmetric* scoring rules (Table 4). Here, $\boldsymbol{\rho}_\theta(x_{1:K}) \triangleq [1, \rho_\theta(x_1), \ldots, \rho_\theta(x_K)] \in \mathbb{R}_+^{K+1}$ and $\rho_\theta(x) \triangleq \frac{r_\theta(x)}{\nu}$. The objective in the first row corresponds to our proposal InfoNCE-anchor. When $K = 1, \nu = 1$, it is also known as JS (Poole et al., 2019) or NT-Logistics (Chen et al., 2020).

| Sym. scoring rule | $\mathcal{L}_{K;\nu}^{\Psi_\Phi}(\boldsymbol{\eta}_\theta)$ (see Eq. (13)) |
|---|---|
| Log | $\frac{K}{K+\nu}\mathbb{E}_{q_1(x_{1:K})}[-\log\frac{\rho_\theta(x_1)}{\|\boldsymbol{\rho}_\theta(x_{1:K})\|_1}] + \frac{1}{K+\nu}\mathbb{E}_{q_0(x_{1:K})}[-\log\frac{\nu}{\|\boldsymbol{\rho}_\theta(x_{1:K})\|_1}]$ |
| Power ($\alpha \notin \{0,1\}$) | $\frac{K}{K+\nu}\mathbb{E}_{q_1(x_{1:K})}[\frac{1}{\alpha}(\frac{\|\boldsymbol{\rho}_\theta(x_{1:K})\|_\alpha}{\|\boldsymbol{\rho}_\theta(x_{1:K})\|_1})^\alpha + \frac{1}{1-\alpha}(\frac{\rho_\theta(x_1)}{\|\boldsymbol{\rho}_\theta(x_{1:K})\|_1})^{\alpha-1}]$ $+ \frac{\nu}{K+\nu}\mathbb{E}_{q_0(x_{1:K})}[\frac{1}{\alpha}(\frac{\|\boldsymbol{\rho}_\theta(x_{1:K})\|_\alpha}{\|\boldsymbol{\rho}_\theta(x_{1:K})\|_1})^\alpha + \frac{1}{1-\alpha}(\frac{1}{\|\boldsymbol{\rho}_\theta(x_{1:K})\|_1})^{\alpha-1}]$ |
| Sym. inverse log | $\frac{K}{K+\nu}\mathbb{E}_{q_1(x_{1:K})}[\frac{\|\log\boldsymbol{\rho}_\theta(x_{1:K})\|}{\|\boldsymbol{\rho}_\theta(x_{1:K})\|_1} + \frac{\|\boldsymbol{\rho}_\theta(x_{1:K})\|_1}{\rho_\theta(x_1)}]$ $+ \frac{\nu}{K+\nu}\mathbb{E}_{q_0(x_{1:K})}[\frac{\log\prod\boldsymbol{\rho}_\theta(x_{1:K})}{\|\boldsymbol{\rho}_\theta(x_{1:K})\|_1} + \|\boldsymbol{\rho}_\theta(x_{1:K})\|_1]$ |
| Pseudo-spherical ($\alpha \notin \{0,1\}$) | $\frac{K}{K+\nu}\mathbb{E}_{q_1(x_{1:K})}[(\frac{\rho_\theta(x_1)}{\|\boldsymbol{\rho}_\theta(x_{1:K})\|_\alpha})^{\alpha-1}] + \frac{\nu}{K+\nu}\mathbb{E}_{q_0(x_{1:K})}[(\frac{1}{\|\boldsymbol{\rho}_\theta(x_{1:K})\|_\alpha})^{\alpha-1}]$ |
| ($\alpha = 2$) | $\frac{K}{K+\nu}\mathbb{E}_{q_1(x_{1:K})}[\frac{\rho_\theta(x_1)}{\|\boldsymbol{\rho}_\theta(x_{1:K})\|_2}] + \frac{\nu}{K+\nu}\mathbb{E}_{q_0(x_{1:K})}[\frac{1}{\|\boldsymbol{\rho}_\theta(x_{1:K})\|_2}]$ |

## G  Deferred Proofs

### G.1  Proof of Proposition 1

*Proof of Proposition 1.* We have an alternative proof for a loose upper bound

$$-\mathcal{L}_{K;0}(\theta) + \log K \leq D(q_1 \| q_0).$$

We first consider the NWJ variational lower bound of the KL divergence:

$$D(q_1 \| q_0) \geq \mathbb{E}_{q_1}[\log r] - \mathbb{E}_{q_0}[r] + 1.$$

Here the equality holds if and only if $r(x) \equiv \frac{q_1(x)}{q_0(x)}$. For $K \geq 2$, consider two distributions $q_1(x_1)q_0(x_2)\cdots q_0(x_K)$ and $q_0(x_1)q_0(x_2)\cdots q_0(x_K)$. Applying the NWJ lower bound, we obtain

$$D(q_1(x) \| q_0(x))$$
$$= D(q_1(x_1)q_0(x_2)\cdots q_0(x_K) \| q_0(x_1)q_0(x_2)\cdots q_0(x_K))$$
$$\geq \mathbb{E}_{q_1(x_1)q_0(x_2)\cdots q_0(x_K)}[\log r(x_1, \ldots, x_K)] - \mathbb{E}_{q_0(x_1)q_0(x_2)\cdots q_0(x_K)}[r(x_1, \ldots, x_K)] + 1.$$

Note that, again, the equality is attained if and only if

$$r(x_1, \ldots, x_K) \equiv \frac{q_1(x_1)}{q_0(x_1)}.$$

Table 7: Examples of InfoNCE-type DRE objectives, derived from *symmetric* scoring rules (Table 4).

| Sym. scoring rule | $\mathcal{L}_{K;0}^{\Psi_\Phi}(\boldsymbol{\eta}_\theta)$ (see Eq. (13)) | Known as |
|---|---|---|
| Log | $\mathbb{E}_{q_1(x_{1:K})}[-\log \frac{r_\theta(x_1)}{\|\mathbf{r}_\theta(x_{1:K})\|_1}]$ | InfoNCE (van den Oord et al., 2018)/NT-Xent (Chen et al., 2020) |
| Power $(\alpha \notin \{0,1\})$ | $\mathbb{E}_{q_1(x_{1:K})}[\frac{1}{\alpha}(\frac{\|\mathbf{r}_\theta(x_{1:K})\|_\alpha}{\|\mathbf{r}_\theta(x_{1:K})\|_1})^\alpha + \frac{1}{1-\alpha}(\frac{r_\theta(x_1)}{\|\mathbf{r}_\theta(x_{1:K})\|_1})^{\alpha-1}]$ | |
| Sym. inverse log | $\mathbb{E}_{q_1(x_{1:K})}[\frac{\log \prod \mathbf{r}_\theta(x_{1:K})}{\|\mathbf{r}_\theta(x_{1:K})\|_1} + \frac{\|\mathbf{r}_\theta(x_{1:K})\|_1}{r_\theta(x_1)}]$ | |
| Pseudo-spherical $(\alpha \notin \{0,1\})$ | $\mathbb{E}_{q_1(x_{1:K})}[(\frac{r_\theta(x_1)}{\|\mathbf{r}_\theta(x_{1:K})\|_\alpha})^{\alpha-1}]$ | |
| $(\alpha = 2)$ | $\mathbb{E}_{q_1(x_{1:K})}[\frac{r_\theta(x_1)}{\|\mathbf{r}_\theta(x_{1:K})\|_2}]$ | |

Now, we consider a specific (suboptimal) parameterization of $r(x_1, \ldots, x_K)$ in the following form:

$$r(x_1, \ldots, x_K) \leftarrow \log \frac{r_\theta(x_1)}{\frac{1}{K}\sum_{k=1}^{K} r_\theta(x_k)}$$

for some nonnegative-valued function $r_\theta \colon \mathcal{X} \to \mathbb{R}_{\geq 0}$. By symmetry, it is easy to show that

$$\mathbb{E}_{q_0(x_1)q_0(x_2)\cdots q_0(x_K)}[r(x_1, \ldots, x_K)] = 1.$$

Hence, the NWJ lower bound simplifies to

$$D(q_1(x) \| q_0(x)) \geq \mathbb{E}_{q_1(x_1)q_0(x_2)\cdots q_0(x_K)}\left[\log \frac{r_\theta(x_1)}{\frac{1}{K}\sum_{k=1}^{K} r_\theta(x_k)}\right] = -\mathcal{L}_{K;0}(\theta), \qquad (14)$$

which concludes the proof. $\square$

### G.2 Proof of Theorem 2

*Proof of Theorem 2.* We start with the following upper bound

$$-\mathcal{L}_{K;\nu}(\theta) = \mathbb{E}_{p(z)p(x_{1:K}|z)}[\log p_\theta(z|x_{1:K})]$$
$$\leq \mathbb{E}_{p(z)p(x_{1:K}|z)}[\log p(z|x_{1:K})],$$

where the upper bound is achieved when $p_\theta(z|x_{1:K}) = p(z|x_{1:K})$. This is by the Gibbs inequality, or equivalently

$$\mathbb{E}_{p(x_{1:K})}\left[D(p(z|x_{1:K}) \| p_\theta(z|x_{1:K}))\right] \geq 0.$$

We note that for $\nu = 0$, we have

$$-\mathcal{L}_{K;0}(\theta) + \log K \leq \mathbb{E}_{p(z)p(x_{1:K}|z)}[\log p(z|x_{1:K})] + \log K$$
$$= \mathbb{E}_{q_1(x_1)q_0(x_2)\cdots q_0(x_K)}\left[\log \frac{\frac{q_1(x_1)}{q_0(x_1)}}{\frac{1}{K}\sum_{z=1}^{K}\frac{q_1(x_z)}{q_0(x_z)}}\right] \qquad (15)$$
$$= D_{\mathsf{JS}}\Big(p(x_{1:K}|z=1), \ldots, p(x_{1:K}|z=K)\Big).$$

The equality condition follows from the Gibbs inequality. This proves the first inequality.

To prove the upper bound $\log K$, continuing from Eq. (15), we have

$$-\mathcal{L}_{K;0}(\theta) + \log K \leq \mathbb{E}_{q_1(x_1)q_0(x_2)\cdots q_0(x_K)}\left[\log \frac{\frac{q_1(x_1)}{q_0(x_1)}}{\frac{1}{K}\sum_{z=1}^{K}\frac{q_1(x_z)}{q_0(x_z)}}\right]$$
$$\leq \mathbb{E}_{q_1(x_1)q_0(x_2)\cdots q_0(x_K)}\left[\log \frac{\frac{q_1(x_1)}{q_0(x_1)}}{\frac{1}{K}\frac{q_1(x_1)}{q_0(x_1)}}\right] = \log K.$$

For the second upper bound, we apply Jensen's inequality with the concavity of the logarithmic function and obtain

$$\mathbb{E}_{q_1(x_1)q_0(x_2)\cdots q_0(x_K)}\left[\log\frac{1}{K}\sum_{z=1}^{K}\frac{q_1(x_z)}{q_0(x_z)}\right] \geq \log\left(\mathbb{E}_{q_1(x_1)q_0(x_2)\cdots q_0(x_K)}\left[\frac{1}{K}\sum_{z=1}^{K}\frac{q_1(x_z)}{q_0(x_z)}\right]\right)$$

$$= \log\left(\frac{1}{K}\chi^2(q_1 \parallel q_0) + 1\right)$$

$$\geq \log\left(\frac{1}{K}(e^{D(q_1 \parallel q_0)} - 1) + 1\right).$$

Here, $\chi^2(p \parallel q) \triangleq \mathbb{E}_p[\frac{p}{q}] - 1$ denotes the *chi-squared divergence* between distributions $p$ and $q$. The last inequality follows since $\chi^2(q_1 \parallel q_0) \geq e^{D(q_1 \parallel q_0)} - 1$. Rearranging the inequality proves the desired bound. $\qquad \square$

### G.3 PROOF OF PROPOSITION 5

To prove this proposition, we need the following lemma. The definition of the generating function $\mathbf{g}^{\boldsymbol{\lambda}}$ of a differentiable loss function $\boldsymbol{\lambda}$ is given in Eq. (11) in Appendix E.1. Recall that the definition of the induced loss function $\boldsymbol{\lambda}^{\Psi}$ for a convex function $\Psi$ is in Eq. (8).

**Lemma 24.** *If $\Psi$ is twice differentiable, $\mathbf{g}^{\boldsymbol{\lambda}^{\Psi}}(\boldsymbol{\eta}) \equiv 0$.*

*Proof of Proposition 5.* By Lemma 24, we have $\mathbf{g}^{\boldsymbol{\lambda}^{\Psi}}(\boldsymbol{\eta}) \equiv 0$. Further, since $\boldsymbol{\eta} \mapsto -f^{\boldsymbol{\lambda}^{\Psi}}(\boldsymbol{\eta}) = \eta_0 \Psi(\boldsymbol{\rho})$ is a perspective of the function $\boldsymbol{\rho} \mapsto \Psi(\boldsymbol{\rho})$, $f^{\boldsymbol{\lambda}^{\Psi}}$ must be (strictly) concave if $\Psi$ is (strictly) convex. Hence, by Theorem 12, we conclude that $\boldsymbol{\lambda}^{\Psi}$ is (strictly) proper. $\qquad \square$

We now prove Lemma 24.

*Proof of Lemma 24.* Consider

$$g_z^{\boldsymbol{\lambda}^{\Psi}}(\boldsymbol{\eta}) = \sum_{z=0}^{M}\eta_z\frac{\partial\lambda_z^{\Psi}(\boldsymbol{\eta})}{\partial\eta_z}.$$

Note that $\rho_{z'} = \eta_{z'}/\eta_0$ for $z' = 1, \ldots, M$, we have

$$\frac{\partial\rho_{z'}}{\partial\eta_z} = \begin{cases} -\frac{\rho_{z'}}{\eta_0} & z = 0 \\ \frac{\mathbb{1}\{z=z'\}}{\eta_0} & z = 1, \ldots, M. \end{cases}$$

**Case 1:** $z = 0$. If $z = 0$, $\lambda_0^{\Psi}(\boldsymbol{\eta}) = \langle\nabla\Psi(\boldsymbol{\rho}), \boldsymbol{\rho}\rangle - \Psi(\boldsymbol{\rho})$. Hence, we have

$$\frac{\partial\lambda_0^{\Psi}(\boldsymbol{\eta})}{\partial\eta_0} = \sum_{z'=1}^{M}\frac{\partial\rho_{z'}}{\partial\eta_z}\frac{\partial}{\partial\rho_{z'}}(\langle\nabla\Psi(\boldsymbol{\rho}), \boldsymbol{\rho}\rangle - \Psi(\boldsymbol{\rho}))$$

$$= \sum_{z'=1}^{M} -\frac{\rho_{z'}}{\eta_0}(\nabla^2\Psi(\boldsymbol{\rho})\boldsymbol{\rho})_{z'}$$

$$= -\frac{1}{\eta_0}(\langle\boldsymbol{\rho}, \nabla^2\Psi(\boldsymbol{\rho})\boldsymbol{\rho}\rangle) - (\nabla^2\Psi(\boldsymbol{\rho})\boldsymbol{\rho})_0). \qquad (16)$$

If $1 \leq z \leq M$, $\lambda_z^{\Psi}(\boldsymbol{\eta}) = -(\nabla\Psi(\boldsymbol{\rho}))_z$, and thus

$$\frac{\partial\lambda_z^{\Psi}(\boldsymbol{\eta})}{\partial\eta_0} = \sum_{z'=1}^{M}\frac{\partial\rho_{z'}}{\partial\eta_z}\frac{\partial}{\partial\rho_{z'}}\left(-\frac{\partial\Psi(\boldsymbol{\rho})}{\partial\rho_z}\right)$$

$$= \frac{1}{\eta_0}\sum_{z'=1}^{M}\rho_{z'}\frac{\partial^2\Psi(\boldsymbol{\rho})}{\partial\rho_z\,\partial\rho_{z'}}$$

$$= \frac{1}{\eta_0}(\nabla^2\Psi(\boldsymbol{\rho})\boldsymbol{\rho})_z. \qquad (17)$$

From (16) and (17), we have

$$g_0^{\boldsymbol{\lambda}^\Psi}(\boldsymbol{\eta}) = \sum_{z=0}^{M} \eta_z \frac{\partial \lambda_z^\Psi(\boldsymbol{\eta})}{\partial \eta_0}$$

$$= -(\langle \boldsymbol{\rho}, \nabla^2 \Psi(\boldsymbol{\rho})\boldsymbol{\rho}\rangle - (\nabla^2\Psi(\boldsymbol{\rho})\boldsymbol{\rho})_0) + \sum_{z=1}^{M} \frac{\eta_z}{\eta_0}(\nabla^2\Psi(\boldsymbol{\rho})\boldsymbol{\rho})_z$$

$$= -\langle \boldsymbol{\rho}, \nabla^2\Psi(\boldsymbol{\rho})\boldsymbol{\rho}\rangle + \sum_{z=0}^{M} \rho_z(\nabla^2\Psi(\boldsymbol{\rho})\boldsymbol{\rho})_z$$

$$= 0.$$

**Case 2:** $1 \leq z \leq M$. If $z = 0$, we have

$$\frac{\partial \lambda_0^\Psi(\boldsymbol{\eta})}{\partial \eta_z} = \frac{1}{\eta_0}(\nabla^2\Psi(\boldsymbol{\rho})\boldsymbol{\rho})_z. \tag{18}$$

If $1 \leq z \leq M$,

$$\frac{\partial \lambda_z^\Psi(\boldsymbol{\eta})}{\partial \eta_z} = -\frac{\partial}{\partial \eta_z}\frac{\partial \Psi(\boldsymbol{\rho})}{\partial \rho_z}$$

$$= -\frac{\partial \rho_z}{\partial \eta_z}\frac{\partial^2\Psi(\boldsymbol{\rho})}{\partial \rho_z \partial \rho_z}$$

$$= -\frac{1}{\eta_0}\frac{\partial^2\Psi(\boldsymbol{\rho})}{\partial \rho_z \partial \rho_z}. \tag{19}$$

Therefore, from (18) and (19), we have

$$g_z^{\boldsymbol{\lambda}^\Psi}(\boldsymbol{\eta}) = \sum_{z=0}^{M} \eta_z \frac{\partial \lambda_z^\Psi(\boldsymbol{\eta})}{\partial \eta_z}$$

$$= (\nabla^2\Psi(\boldsymbol{\rho})\boldsymbol{\rho})_z - \sum_{z=1}^{M} \rho_z \frac{\partial^2\Psi(\boldsymbol{\rho})}{\partial \rho_z \partial \rho_z}$$

$$= (\nabla^2\Psi(\boldsymbol{\rho})\boldsymbol{\rho})_z - (\nabla^2\Psi(\boldsymbol{\rho})\boldsymbol{\rho})_z$$

$$= 0.$$

Hence, we conclude that $\mathbf{g}^{\boldsymbol{\lambda}^\Psi}(\boldsymbol{\eta}) \equiv 0$. $\qquad\square$

### G.4 PROOF OF THEOREM 6

We note that, while the following proof is self-contained, a more detailed technical discussion on the general relationship between proper scoring rule and Bregman divergence minimization in Appendix E.4.

*Proof of Theorem 6.* Note that we can write

$$\mathcal{L}_{K;\nu}^\Psi(\boldsymbol{\eta}_\theta) - \mathcal{L}_{K;\nu}^\Psi(\boldsymbol{\eta}^*) = \mathbb{E}_{p(x_{1:K})}\Big[\langle \boldsymbol{\eta}^*(x_{1:K}), \boldsymbol{\lambda}^\Psi(\boldsymbol{\eta}_\theta(x_{1:K}))\rangle - \langle \boldsymbol{\eta}^*(x_{1:K}), \boldsymbol{\lambda}^\Psi(\boldsymbol{\eta}^*(x_{1:K}))\rangle\Big].$$

Now, it is easy to check that, we have

$$\langle \boldsymbol{\eta}^*, \boldsymbol{\lambda}^\Psi(\boldsymbol{\eta}_\theta)\rangle = \eta_0^*\Big(-\Psi(\boldsymbol{\rho}_\theta) - \langle \nabla_{\boldsymbol{\rho}}\Psi(\boldsymbol{\rho}_\theta), \boldsymbol{\rho}^* - \boldsymbol{\rho}_\theta\rangle\Big).$$

In particular,

$$\langle \boldsymbol{\eta}^*, \boldsymbol{\lambda}^\Psi(\boldsymbol{\eta}^*)\rangle = -\eta_0^*\Psi(\boldsymbol{\rho}^*).$$

Hence, we have

$$\langle \boldsymbol{\eta}^*, \boldsymbol{\lambda}^\Psi(\boldsymbol{\eta}_\theta)\rangle - \langle \boldsymbol{\eta}^*, \boldsymbol{\lambda}^\Psi(\boldsymbol{\eta}^*)\rangle = \eta_0^* B_\Psi(\boldsymbol{\rho}^*, \boldsymbol{\rho}_\theta).$$

From this expression, we have

$$\mathcal{L}_{K;\nu}^{\Psi}(\boldsymbol{\eta}_\theta) - \mathcal{L}_{K;\nu}^{\Psi}(\boldsymbol{\eta}^*) = \mathbb{E}_{p(x_{1:K})}\left[\eta_0^* B_\Psi(\boldsymbol{\rho}^*, \boldsymbol{\rho}_\theta)\right]$$

$$= \mathbb{E}_{p(x_{1:K})}\left[p(z=0|x_{1:K})B_\Psi(\boldsymbol{\rho}^*, \boldsymbol{\rho}_\theta)\right]$$

$$= p(z=0)\mathbb{E}_{p(x_{1:K}|z=0)}\left[B_\Psi(\boldsymbol{\rho}^*, \boldsymbol{\rho}_\theta)\right].$$

Since $p(z=0) = \frac{\nu}{K+\nu}$ and $p(x_{1:K}|z=0) = q_0(x_1)q_0(x_2)\cdots q_0(x_K)$ by definition, this concludes the proof. □

## H  EXPERIMENT DETAILS

This section provides the details on the experiments in the main text. All implementations are based on PyTorch and all experiments were performed on a single NVIDIA GeForce RTX 3090. Our code to reproduce the results is made publicly available at `https://github.com/jongharyu/infonce-anchor`.

### H.1  MI ESTIMATION

We conducted a series of mutual information (MI) estimation experiments across three distinct data modalities: synthetic Gaussian variables, image-based representations from MNIST, and text embeddings derived from the IMDB dataset, using the standardized mibenchmark framework (Lee & Rhee, 2024). Each experiment paired a 10-dimensional synthetic source variable $X \in \mathbb{R}^{10}$ with a modality-specific target variable $Y$, varying in dimensionality depending on the data type. Across all experiments, we used a consistent training configuration: models were optimized using Adam with a learning rate of 1e-4, trained in stepwise mode for 20,000 iterations.

Across all setups, we evaluated a fixed set of mutual information estimators, including NWJ, NWJ-Plugin, JS, JS-Plugin, InfoNCE, InfoNCE-Anchor, Density Ratio Fitting, and Spherical, with both joint and separable critic types. The critic network in all cases was an MLP composed of two hidden layers with 512 units, ReLU activations, and no normalization or dropout layers. Critic architectures projected inputs into a shared 16-dimensional embedding space. For joint critics, $X$ and $Y$ pairs were concatenated and passed through a single encoder, whereas for separable critics, independent encoders $g(x)$ and $h(y)$ were used. We used batch size 64 for all experiments except the ones in Figure 3. We refer the readers to (Lee & Rhee, 2024) and their codebase for the rest of the details including the data generation mechanism.

Figure 3 summarizes the result of MI estimation for the Gaussian experiment with cubic transformation with varying batch sizes. It clearly shows that InfoNCE-anchor exhibits a consistent performance, but we note that $\mathsf{JS}_{\mathsf{plugin}}$ also performs remarkably well in this simple benchmark.

### H.2  PROTEIN INTERACTION PREDICTION

We followed the same setup of Gowri et al. (2024), and here we briefly overview the essential part. We conducted experiments on two datasets derived from ProtTrans5-encoded protein embeddings: one composed of 22,229 kinase–target pairs and another with 1,702 ligand–receptor pairs. Each protein is represented by a 1,024-dimensional vector, and all embeddings were whitened and clipped to the range $[-10, 10]$. Across 20 trials, 170 proteins were randomly selected and held out per trial, ensuring that no interaction in the training set included any of the held-out proteins. The remaining interactions were used for training a mutual information estimator.

Our approach trains a separable critic network to estimate the density ratio via the InfoNCE-anchor objective. The critic architecture is a MLP with 4 hidden layers, each containing 256 units, and outputs 32-dimensional embeddings for each input protein, separate encoders $f(x)$ and $g(y)$ for each side of the pair. ReLU activation was used, and no normalization layers were applied by default. We used the Adam optimizer with a learning rate of 1e-4, batch size of 64, and 10,000 training steps. We implement early stopping with a patience of 500 steps, based on validation loss, which is monitored every 500 iterations. The final model is selected based on the best validation performance and is then used to estimate pointwise mutual information (PMI) for held-out protein pairs.

We present ROC curves (Figure 4) and histograms of learned PMI values (Figure 5) for each estimator. These two figures clearly demonstrate that InfoNCE-anchor exhibit the best discriminative power.

## H.3 SELF-SUPERVISED REPRESENTATION LEARNING

Here, we provide details on the objective functions we considered in the experiment. We used the temperature parameter $\tau = 0.2$ throughout, unless stated otherwise.

- InfoNCE: Log score, $K = B - 1$, $\nu = 0$, PMI factorization.
- InfoNCE-anchor: Log score, $K = B - 1$, $\nu = 1$, PMI factorization.
- JS: Log score, $K = 1$, $\nu = 1$, PMI factorization.
- Spherical: Spherical score, $K = B - 1$, $\nu = 1$, PD factorization.
- $\chi^2$: Asymmetric power score with $\alpha = 2$, $K = 1$, $\nu = 1$. In this case, $\tau = 0.1$ was used.

We found that the PMI factorization was not effective for all scoring rules other than the log score. The rest of the experimental details can be found from the codebase of da Costa et al. (2022).

To confirm that the observation is not specific to the ResNet backbone, we conducted additional representation learning experiments using two different ViT-base and ViT-small backbones; see Table 8. The reported numbers are top-1 accuracies on the validation set. These results support a consistent conclusion that the anchor does not improve the quality of learned representation. In the revision, we will also add experiments on ImageNet to verify the trend.

Table 8: Linear probing top-1 accuracy (%) on CIFAR-100 across different backbones.

| Backbone | InfoNCE | InfoNCE-anchor | |
| --- | --- | --- | --- |
| | | $\nu = 1$ | $\nu = 1024$ |
| ResNet-18 | 65.98 | 65.74 | – |
| ViT-small | 63.35 | 62.33 | 62.32 |
| ViT-base | 67.57 | 65.22 | 65.15 |

## H.4 IMPACT OF THE ANCHOR PARAMETER $\nu$

In the experiments in Section 4, we assumed $\nu = 1$ by default. This setting is robust and consistently outperforms standard InfoNCE without any tuning. However, as $\nu$ represents the relative weight of the anchor class, we analyze its impact to provide practical guidance.

**Stability in MI Estimation.** As shown in Figure 6, the estimator is highly stable across a wide log-scale range of $\nu$. While the optimal $\nu$ varies by dataset (e.g., higher for Gaussian, lower for Text), the estimation error remains low throughout the sweep. This confirms that the method is not fragile to hyperparameter selection.

**Gains in Downstream Tasks.** Table 9 shows that specific tasks benefit from larger $\nu$ values. For Kinase prediction, increasing $\nu$ improves the AUROC from $0.81$ to $0.85$. This demonstrates that better MI estimation through the anchor modification directly translates to improved performance in specialized downstream applications.

Table 9: Protein Interaction Prediction performance (AUROC) for varying anchor parameter $\nu$.

| Dataset | $\nu = 2^0$ | $\nu = 2^2$ | $\nu = 2^4$ | $\nu = 2^6$ | $\nu = 2^8$ |
| --- | --- | --- | --- | --- | --- |
| Ligand | $0.97_{\pm.013}$ | $0.97_{\pm.013}$ | $0.98_{\pm.012}$ | $0.98_{\pm.009}$ | $\mathbf{0.98_{\pm.009}}$ |
| Kinase | $0.81_{\pm.063}$ | $0.83_{\pm.052}$ | $0.82_{\pm.050}$ | $\mathbf{0.85_{\pm.052}}$ | $0.84_{\pm.048}$ |

**Practical Guidance.** We recommend $\nu = 1$ as a robust starting point, but cross-validation with a coarse log-scale sweep (e.g., $2^0$ to $2^{10}$) can be performed for a fine-grained tuning. In our SSL experiments, linear probe accuracy remained stable regardless of $\nu$, suggesting that the anchor's primary benefit is for tasks requiring precise density ratios.

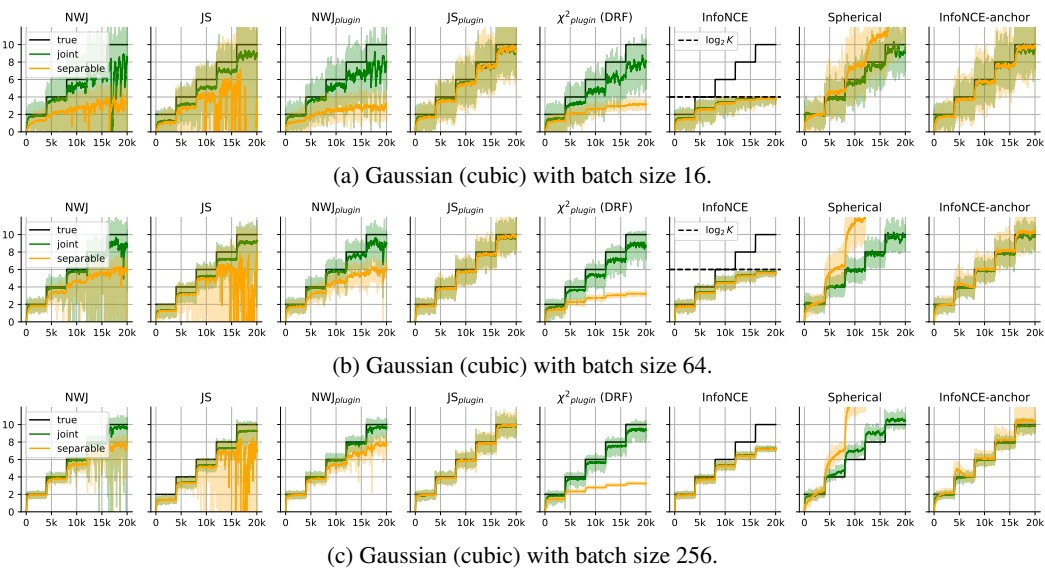

Figure 3: Summary of MI estimation results on the standard benchmark on the Gaussian cubic data, with different batch sizes.

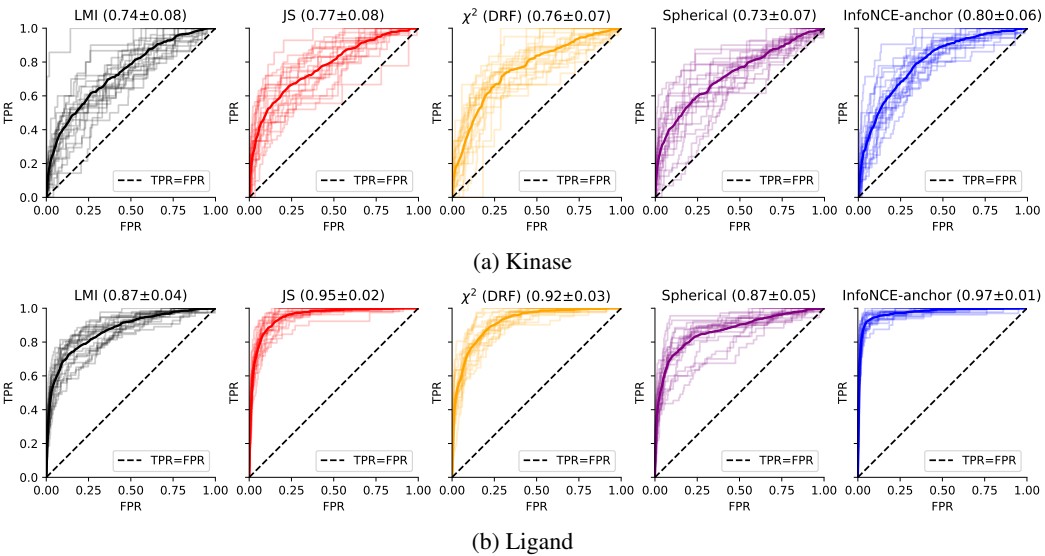

Figure 4: ROC curves from different estimators.

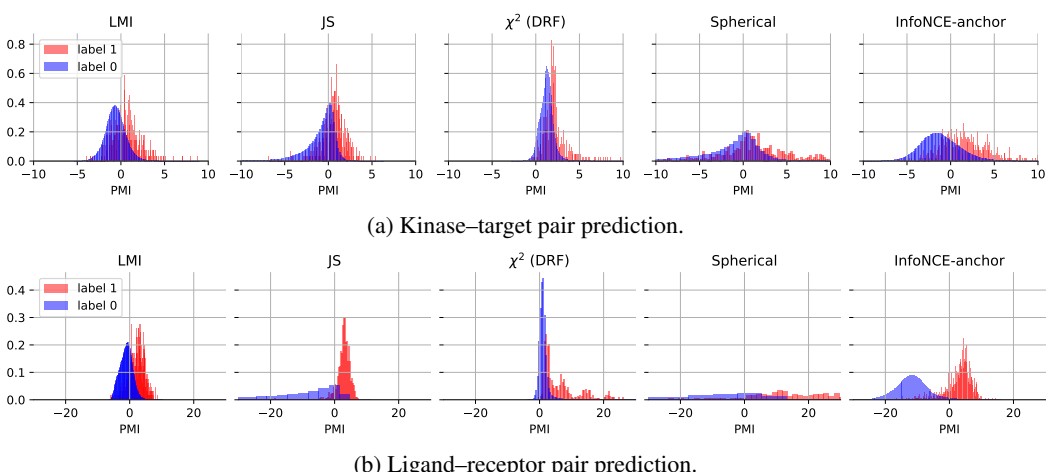

(a) Kinase–target pair prediction.

(b) Ligand–receptor pair prediction.

Figure 5: Histograms of pointwise MI ($\log \frac{p(x,y)}{p(x)p(y)}$) from different estimators.

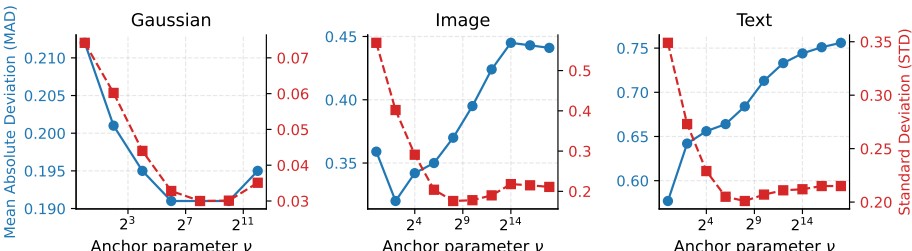

Figure 6: Ablation study on the anchor parameter $\nu$ across different datasets for MI estimation. The estimator remains stable across a wide range of $\nu$, with optimal values being dataset-dependent.

