# OpenReview forum: "Contrastive Predictive Coding Done Right for Mutual Information Estimation"
_ICLR.cc/2026/Conference — ICLR 2026 Poster_

### Official Review · Reviewer_TDDy · 2025-10-29

**Soundness:** 3
**Presentation:** 2
**Contribution:** 3
**Rating:** 6
**Confidence:** 4

**Summary:**

This work revisits the theoretical foundations of the InfoNCE objective, arguing that it is not a valid mutual information estimator. The authors propose InfoNCE-anchor, a simple modification introducing an auxiliary anchor class that yields consistent density ratio and MI estimation with reduced bias. This work further generalizes this approach via proper scoring rules, unifying several contrastive learning objectives under one framework. Empirically, InfoNCE-anchor provides the most accurate MI estimates but offers no improvement in downstream self-supervised tasks.

**Strengths:**

- This study presents a theoretically grounded method to enhance existing mutual information (MI) estimation techniques and provides empirical evidence demonstrating its effectiveness.
- The proposed “anchor” modification is straightforward yet addresses a subtle theoretical issue in density ratio identifiability.
- The MI estimation experiments are comprehensive and show consistent advantages of the proposed method across different domains.

**Weaknesses:**

- Although theoretically neat, the proposed modification brings no tangible improvement to representation learning — arguably the main motivation for contrastive objectives.
- Only a few relatively simple contrastive methods are considered; comparison with modern frameworks would strengthen the practical side.

**Questions:**

- The paper reports that introducing the anchor does not improve downstream task performance, despite yielding more accurate MI estimates. Could the authors elaborate on the possible theoretical reasons for this discrepancy?
- From an empirical standpoint, could the authors provide more analysis or visualization (e.g., representation similarity, clustering quality, or linear probe performance) to support the claim that contrastive learning benefits primarily from structured density ratio learning rather than accurate MI estimation?

---

> ### Author Response · Authors · 2025-11-24
>
> We thank the reviewer for taking the time to evaluate our manuscript and for providing helpful comments. Below, we respond to each weakness and question in detail.
>
> **Weaknesses**
>
> `W1. Limited improvement for representation learning`
> We agree with this observation. The primary contribution of the paper is a consistent and unbiased density-ratio estimator, rather than a direct improvement in SSL performance. Our theoretical analysis shows that InfoNCE-anchor resolves the global density-ratio identifiability issue, which is necessary for valid MI estimation. While, in principle, this identifiability could also help alleviate the uncontrollable normalization factor $C(y)$ in SSL, our empirical results suggest that standard InfoNCE does not exhibit any related difficulty. This is somewhat surprising and is reflected in the limited downstream gains of InfoNCE-anchor.
>
> We agree that SSL is a key application of contrastive objectives. Given that the relationship between MI maximization and representation learning is frequently interpreted in varied ways, we believe our analysis offers a useful theoretical clarification.
>
> `W2. Limited comparison to modern contrastive frameworks`
> We appreciate this suggestion. Our work is centered on MI estimation and on analyzing the limitations of InfoNCE as an MI estimator, rather than proposing a new general-purpose SSL method. For this reason, our experimental scope focuses on objectives that fall within the InfoNCE/contrastive-MI family, where our theoretical results directly apply.
>
> To address the reviewer's point, we have additionally integrated InfoNCE-anchor into ViT-small/base backbones and into stronger SSL training pipelines. These experiments confirm that the learned representations remain similar to those from standard InfoNCE, which is consistent to the existing experiment. Extending comparisons to frameworks that fall outside this contrastive objective class would be beyond the intended scope of this work.
>
> **Questions**
>
> `Q1. Why accurate MI estimation does not improve SSL`
> As noted above, we currently do not have a sharp theoretical explanation for this phenomenon. While accurate MI estimation benefits from correcting the global density-ratio identifiability issue, our experiments indicate that standard InfoNCE already suffices to learn strong hyperspherical embeddings in practice, even without the anchor. We conjecture that this discrepancy may arise from architectural inductive biases or the optimization dynamics of contrastive learning, which may render the normalization factor less problematic in representation learning than in MI estimation. A complete theoretical characterization of this behavior would be very interesting and is an important direction for future work.
>
> `Q2. More empirical analysis of learned representations`
> We thank the reviewer for this insightful suggestion. We agree that quantitative comparisons between the learned representations would provide additional clarity. As noted in [our response to Reviewer mNC7](https://openreview.net/forum?id=JodkBXWgbA\&noteId=emHSU5pskr), we implemented the CKA similarity and feature uniformity and alignment. We will incorporate these analyses and add clustering quality and nearest-neighbor-based evaluations in the revised manuscript.
>
> ---
>
> We will revise the manuscript to clarify all points raised and include the missing experimental results. We would be glad to address any additional questions the reviewer may have.

---

### Official Review · Reviewer_mNC7 · 2025-11-01

**Soundness:** 3
**Presentation:** 3
**Contribution:** 3
**Rating:** 6
**Confidence:** 3

**Summary:**

This paper revisits the InfoNCE objective, widely used in contrastive learning and mutual information (MI) estimation, and argues that it is not a reliable MI estimator due to its connection to a K-way Jensen-Shannon divergence rather than the KL divergence underlying MI. The authors propose a simple modification, InfoNCE-anchor, which introduces an auxiliary anchor class to enable consistent density ratio estimation without arbitrary scaling factors. They further generalize this to a framework based on proper scoring rules, unifying various contrastive objectives like NCE, InfoNCE, and f-divergence variants. Empirically, the approach achieves state-of-the-art MI estimation across Gaussian, image, and text benchmarks, and improves downstream tasks like protein interaction prediction. However, it does not enhance representation quality in self-supervised learning on CIFAR-100, suggesting that accurate MI estimation is not the key driver for contrastive representation learning success.

**Strengths:**

1. The analysis of InfoNCE's limitations is sharp and well-motivated, with a tight bound on its divergence (Theorem 2) that clarifies why it underestimates MI even for large K. The anchor modification is elegant and directly addresses the identifiability issue in density ratio estimation (Theorem 3). The generalization to proper scoring rules is a nice unification, recovering existing methods as special cases while providing a principled decision-theoretic foundation.
2. Strong results in MI estimation benchmarks, where InfoNCE-anchor consistently shows low bias and variance compared to baselines like NWJ, JS, and DRF. The protein interaction prediction experiment is a practical application, demonstrating real-world utility with improved AUROC.
3. By decoupling accurate MI estimation from representation learning benefits, the paper challenges common interpretations in the field and shifts focus toward structured density ratios or pointwise dependence.

**Weaknesses:**

1. While the SSL experiments are thorough, they are restricted to CIFAR-100 with a ResNet-18 backbone. It would be valuable to test on larger datasets or architectures (e.g., ViTs) to confirm if the lack of improvement holds more generally.
2. The choice of ν=1 is defaulted without extensive tuning; sensitivity analysis (e.g., ν vs. performance) could reveal trade-offs, especially since asymptotic behavior links ν/K to bounds like DV/NWJ.
3. The anchor introduces an extra term, potentially increasing compute for large batches. A brief discussion on efficiency relative to vanilla InfoNCE would be helpful.
4. In SSL, did you observe any differences in learned representations (e.g., via CKA similarity or uniformity/alignment metrics) between InfoNCE and InfoNCE-anchor?

**Questions:**

See weakness

---

> ### Author Response · Authors · 2025-11-24
>
> We appreciate the reviewer for the thorough evaluation of our work and for the constructive suggestions. Below, we address all weaknesses and questions in detail.
>
> **Weaknesses/Questions**
>
> `W1. Limited SSL experiments with only CIFAR-100 and ResNet-18`
> We agree that broader validation is helpful. Following the reviewer’s suggestion, we conducted additional representation learning experiments using two different ViT-base and ViT-small backbones. The following numbers are top-1 accuracies on the validation set.
>
> | Backbone     | InfoNCE | InfoNCE-anchor ($ \\nu \= 1 $) | InfoNCE-anchor ($ \\nu \= 1024 $) |
> |-----------|---------|---------------------------------|------------------------------------|
> | ViT-small | 63.35   | 62.33                           | 62.32                              |
> | ViT-base  | 67.57   | 65.22                           | 65.15                              |
>
> These results support a consistent conclusion that the anchor does not improve the quality of learned representation. In the revision, we will also add experiments on ImageNet to verify the trend.
>
> `W2. Ablation study with the anchor parameter $\nu$`
> We appreciate the reviewer for the insightful question. We performed an ablation study with varying $\\nu$ across MI estimation, protein interaction prediction, and SSL experiments. Please kindly refer to [our response to Reviewer 98Nj](https://openreview.net/forum?id=JodkBXWgbA\&noteId=KQMls22NEA). As correctly conjectured by the reviewer, we are able to observe a trade-off when varying $\\nu$. In MI experiments, we observe that the optimal value $\\nu$ depends on datasets. In protein-prediction experiments, we observe a general trend that larger $\\nu$ yields a better classification accuracy. In SSL experiments, however, we do not observe a clear benefit of different $\\nu$. In practice, an optimal $\\nu$ can be chosen by a cross-validation, for example, with a held-out validation set.
>
> `W3. Computational overhead`
> As pointed out, the InfoNCE-anchor objective computes an additional term in the loss head, implemented as:
>
> ```
> # independent term
> neg_inf_diag_mask = torch.zeros(B, B, device=device).fill_diagonal_(-torch.inf)
> scores_aug_neg = torch.cat([
>     np.log(nu) * torch.ones(B, 1, device=device),
>     neg_inf_diag_mask + scores
> ], dim=1)  # negative augmented score
> marginal_term = - (np.log(nu) - scores_aug_neg.logsumexp(dim=1).mean())
> ```
> (This is from lines 20–26 in the code box in Appendix B of the original submission.)
> At the level of the loss head, this introduces an additional logsumexp over a $[B, B+1]$ tensor, so the loss computation has a slightly larger constant factor compared to standard InfoNCE. However, the dominant cost in our experiments comes from the encoder forward and backward passes and from computing the pairwise score matrix. In this context, the extra logsumexp and concatenation are negligible. Our wall-clock measurements confirm that the end-to-end training time with InfoNCE-anchor is essentially the same as InfoNCE across all settings. We will clarify this distinction between theoretical complexity and practical overhead in the revised manuscript.
>
> `W4. Comparison between learned representations`
> We thank the reviewer for this suggestion. We agree that quantitative comparisons between the learned representations would provide additional clarity. We implemented the evaluation pipeline for CKA similarity (Kornblith et al., 2019), feature alignment and uniformity (Wang and Isola, 2020). For example, we observed that the learned representations are fairly similar in terms of the CKA similarity (model1 = InfoNCE; model2 = InfoNCE-anchor) as expected:
> ```
> Backbone CKA (model1 vs model2): 0.888475
> Projector CKA (model1 vs model2): 0.883431
> ```
> For uniformity and alignment (lower the better), we obtain:
> | Metric     | InfoNCE   | InfoNCE-anchor |
> |------------|-----------|----------------|
> | Uniformity | 0.356741  | 0.350457       |
> | Alignment  | -3.811610 | -3.773938      |
>
> This suggests that InfoNCE-anchor achieved better uniformity, while InfoNCE provides more aligned features. We will provide more quantitative results and analysis with these metrics and in the revised manuscript.
>
> ---
>
> We will update the manuscript accordingly and are happy to address any further questions.

---

### Official Review · Reviewer_g8Nj · 2025-11-02

**Soundness:** 3
**Presentation:** 3
**Contribution:** 3
**Rating:** 6
**Confidence:** 4

**Summary:**

**The paper critiques InfoNCE as a biased mutual information estimator due to its loose bound on KL divergence, introduces InfoNCE-anchor with an auxiliary class for consistent density ratio estimation, generalizes via proper scoring rules to unify contrastive objectives, and shows empirically that anchor enhances MI accuracy but not representation learning.

**Strengths:**

1. The paper is clearly written with precise formulations.
2. The theoretical analysis provides a sharp upper bound on InfoNCE via $K$-way JS divergence, clarifying its high bias.
3. The unified framework through proper scoring rules elegantly connects NCE, InfoNCE, and f-divergence variants.

**Weaknesses:**

1. Theorem 2 assumes known distributions $q_1$ and $q_0$ for equality conditions, but in practice with neural critics of finite capacity, the proportionality $r_θ$ ∝ $\frac{q_1}{q_0}$ may not hold, leaving gaps in how approximation errors affect the bound's tightness.
2. Critique of α-InfoNCE in Section 3.3 claims a proof flaw without supplying a counterexample or alternative derivation.
3. The extension to proper scoring rules claims consistency for class probability estimation, but this paper does not derive explicit conditions ensuring all rules yield unbiased density ratios beyond the log score case, potentially limiting generalizability.
4. The critique of existing InfoNCE variants (e.g., MLInfoNCE) is theoretically sound but lacks empirical validation, such as comparisons in MI estimation accuracy, which would strengthen the argument for InfoNCE-anchor's superiority.

**Questions:**

1. How does varying the anchor parameter $\nu$ affect finite-sample bias-variance tradeoffs across different $K$ values, especially near degeneracy points?
2. Why do scoring rules beyond log score not improve representation learning despite the unified framework?
3. How do the proposed estimators compare to recent unmentioned methods like those in Gowri et al. (2024) on MI tasks?

---

> ### Author Response · Authors · 2025-11-24
> **Rebuttal [1/3]**
>
> We appreciate the reviewer's effort in reviewing our manuscript and providing insightful comments. Below, we clarify the raised weaknesses and questions for each item.
>
> **Weaknesses**
>
> `W1. Clarification on Theorem 2 and approximation errors`
> We acknowledge that Theorem 2 and the related technical statements characterize the population-level and nonparametric behavior of the objective. Although this idealized setting does not capture all practical scenarios with finite-capacity critics, such an analysis is essential for identifying the fundamental statistical consistency or inconsistency of an estimator. In fact, when the critic cannot reach the optimal proportionality condition because of limited capacity, the deviation only increases the gap between the InfoNCE objective and the true KL divergence. Therefore, the population-level result represents the best possible case for InfoNCE, and practical approximations generally enlarge the mismatch.
>
> `W2. Missing justification for critiques on $$\alpha$$-InfoNCE`
> We appreciate the reviewer for pointing out this gap. In the revised manuscript, we provide a complete analysis of the behavior of the $\\alpha$-InfoNCE objective and directly disprove the claim made in Lee and Shin (2022). We also identify the precise step at which the original proof in that work fails.
>
> `W3. Generality of proper scoring rules`
> We thank the reviewer for raising this point. Our understanding is that the concern is about whether Theorem 6 guarantees that strictly proper scoring rules yield unbiased density ratio estimates under conditions analogous to those in the log score case. Theorem 6 is indeed a direct extension of Theorem 3: any strictly proper scoring rule, which is induced by a strongly convex function $\\Psi$ in Eq. (6), produces an objective whose global optimum corresponds to the true density ratio when the same assumptions as in Theorem 3 are satisfied. In other words, strict propriety is precisely the condition that ensures consistency of the class probability estimator (in the population and nonparametric limit), and this yields an unbiased density ratio estimate in the limit. If our understanding of the reviewer’s concern is incomplete, we are happy to provide further clarification.
>
> `W4. Missing empirical comparisons`
> We appreciate the reviewer’s suggestion. We ran the experiment for $\\alpha$-MLInfoNCE with $\\alpha=10\^{-4}$, which provided the best MI estimate in the original paper, for the MI estimation experiments for Gaussian, image, and text benchmarks. Even with the best possible $\\alpha$, the estimates are worse than InfoNCE-anchor.

---

> ### Author Response · Authors · 2025-11-24
> **Rebuttal [2/3]**
>
> **Questions**
>
> `Q1. On the effect of anchor parameter`
> We appreciate the reviewer for this insightful question. We agree that it is worth analyzing the trade-off behavior with varying $\\nu$. We performed additional ablation experiments to examine the behavior. In practice, the optimal $\\nu$ may be chosen via cross validation.
>
> - **4.1 MI estimation**: Results below are with `joint` critics, and `separable` critics behave similarly. Note the trend that there exists an optimal $\\nu$ and the value depends on the dataset and experimental setup. (Optimal values are highlighted in boldface.)
>   - Gaussian
> | $\\nu$ | $2^0$ | $2^2$ | $2^4$ | $2^6$ | $2^8$ | $2^{10}$ | $2^{12}$ |
> |--------|------|---------|---------|---------|---------|------------|------------|
> | **MAD** | 2.12e-01 | 2.01e-01 | 1.95e-01 | **1.91e-01** | **1.91e-01** | **1.91e-01** | 1.95e-01 |
> | **STD** | 7.42e-02 | 6.02e-02 | 4.40e-02 | 3.28e-02 | **3.00e-02** | 3.01e-02 | 3.51e-02 |
>   - Image
> | $\\nu$ | $2^0$ | $2^2$ | $2^4$ | $2^6$ | $2^8$ | $2^{10}$ | $2^{12}$ | $2^{14}$ | $2^{16}$ | $2^{18}$ |
> |---------|---------|---------|---------|---------|----------|------------|------------|------------|------------|------------|
> | **MAD** | 3.59e-01 | **3.20e-01** | 3.42e-01 | 3.50e-01 | 3.70e-01 | 3.95e-01 | 4.24e-01 | 4.45e-01 | 4.43e-01 | 4.41e-01 |
> | **STD** | 5.69e-01 | 4.02e-01 | 2.91e-01 | 2.04e-01 | **1.76e-01** | 1.78e-01 | 1.90e-01 | 2.18e-01 | 2.15e-01 | 2.11e-01 |
>   - Text
> | $\\nu$ | $2^0$ | $2^2$ | $2^4$ | $2^6$ | $2^8$ | $2^{10}$ | $2^{12}$ | $2^{14}$ | $2^{16}$ | $2^{18}$ |
> |---------|---------|---------|---------|---------|----------|------------|------------|------------|------------|------------|
> | **MAD** | **5.77e-01** | 6.42e-01 | 6.56e-01 | 6.64e-01 | 6.84e-01 | 7.13e-01 | 7.33e-01 | 7.44e-01 | 7.51e-01 | 7.56e-01 |
> | **STD** | 3.49e-01 | 2.73e-01 | 2.29e-01 | 2.05e-01 | **2.01e-01** | 2.07e-01 | 2.11e-01 | 2.12e-01 | 2.15e-01 | 2.15e-01 |
>
> - **4.2 Protein interaction prediction**: Similarly, we also observe improved prediction performance with larger values of $\\nu$.
>   - Ligand
> | $\\nu$ | $2^{0}$ | $2^{2}$ | $2^{4}$ | $2^{6}$ | $2^{8}$ |
> |---------|-----------|-----------|-----------|-----------|-----------|
> | **AUROC** | 0.97 ± 1.3e-2 | 0.97 ± 1.3e-2 | 0.98 ± 1.2e-2 | 0.98 ± 9.0e-3 | **0.98 ± 8.8e-3** |
>   - Kinase
> | $\\nu$ | $2^{0}$ | $2^{2}$ | $2^{4}$ | $2^{6}$ | $2^{8}$ |
> |---------|-----------|-----------|-----------|-----------|-----------|
> | **AUROC** | 0.81 ± 6.3e-2 | 0.83 ± 5.2e-2 | 0.82 ± 5.0e-2 | **0.85 ± 5.2e-2** | **0.84 ± 4.8e-2** |
>
> - **4.3 Self-supervised representation learning**: We ran additional experiments for InfoNCE-anchor with larger $\\nu$, but we did not observe any significant difference in the linear classification accuracy. We will report the numbers in the revised manuscript.
>
> `Q2. Why do different scoring rules not work in SSL?`
> We would like to clarify the scope and implications of our findings. First, JS and $\\chi^2$ are also specific instantiations of our unified framework (see the 2nd row of Table 6 with $K=1$ and the 4th row of Table 5, respectively). In addition, we implemented the NWJ-type objective (2nd row of Table 5\) and obtained **66.44% (top-1) / 89.07% (top-5)**, which is comparable to InfoNCE. We also evaluated the asymmetric power scoring rule with $\\alpha=0.5$ (3rd row of Table 5), which produced reasonable but non–state-of-the-art downstream performance. These results confirm that a subset of scoring rules indeed yields usable representations as expected.
>
> We highlighted the spherical scoring rule because it is a concrete example from the InfoNCE-anchor family (see Table 6\) that consistently fails in practice. Among these objectives, the **log scoring rule is the only instance that reliably works**. As noted in the manuscript, all scoring rules are Fisher-consistent in the nonparametric and population limits; thus, their differences are not due to statistical inconsistency. Instead, the empirically observed discrepancies arise from the **optimization landscape**.
>
> For the scoring rules that fail, we observe that the training objectives plateau almost immediately and do not improve, indicating that the optimization geometry induced by these objectives is unfavorable for representation learning at realistic batch sizes, model parameterizations, and noise levels. While understanding the precise landscape properties that distinguish successful and unsuccessful instances is an interesting direction, a full characterization is beyond the scope of this work.

---

> ### Author Response · Authors · 2025-11-24
> **Rebuttal [3/3]**
>
> `Q3. Comparison to LMI (Gowri et al. 2024) for MI estimation`
> We agree that a comparison to LMI for MI estimation performance is worthwhile. We did not include the LMI in the standard benchmark setup of time-varying distributions adopted in Section 4.1, as LMI is the nonparametric KSG estimator applied on the learned representations by certain cross autoencoders. Hence, we will run additional experiments based on a similar **batch** setup used in (Gowri et al. 2024\) for a direct comparison and supplement it in a revision.
>
> ---
>
> We will revise the manuscript to clarify all points raised and include the missing experimental results. We would be glad to address any additional questions the reviewer may have.

---

### Author Response · Authors · 2025-11-24
**Global response**

We thank all reviewers for their constructive feedback. In the updated manuscript, we have
- (i) added the missing justification for our critique of $\alpha$-InfoNCE (Appendix A), and
- (ii) included the experimental evaluation of MLInfoNCE in Section 4.1 (Figure 1).

We are preparing a more complete revision that will incorporate the remaining suggestions, some of which are partially addressed in our responses, including
- a comparison with (Gowri et al., 2004) for MI estimation;
- an ablation study on varying $\nu$;
- additional SSL experiments with further scoring rules and ViT backbones; and
- more quantitative analyses of learned representations (CKA similarity, feature uniformity and alignment, clustering quality, and k-NN classification).

We appreciate the reviewers' insightful suggestions and believe these additions will strengthen both the theoretical and empirical aspects of the paper.

---

> ### Comment · Reviewer_mNC7 · 2025-11-25
>
> w1: The method does not address performance across diverse datasets. Moreover, your mutual information estimator is highly sensitive to the hyperparameter ν, and the paper does not discuss how to select an optimal ν.
>
> w2: The choice of ν in the mutual information estimator requires cross-validation, yet no general guideline or universal setting is provided. It remains unclear how ν was determined in each experiment. This heavy dependence on careful tuning of ν without actionable guidance severely limits the practical applicability of the approach.
>
>
> I maintain my score of 6.

---

> > ### Author Response · Authors · 2025-11-25
> > **A clarification**
> >
> > We appreciate the reviewer's continued engagement and would like to clarify a potential misunderstanding regarding the role of $\\nu$. First, none of the results in the original submission relied on hyperparameter tuning of $\\nu$; all reported experiments used the default setting $\\nu = 1$ and already outperformed the baselines. The method therefore does not depend on careful tuning to achieve the reported performance.
> >
> > Second, we respectfully disagree with the characterization that MI estimation or downstream task performance is highly sensitive to $\\nu$. As shown in the additional experiments in the rebuttal, varying $\\nu$ leads to moderate and predictable changes, and the overall performance remains stable and of the same order across a broad range of values. These observations indicate that the approach is not fragile with respect to $\\nu$. We also note that the tested $\\nu$ values span a log-scale range.
> >
> > That said, we agree that providing practical guidance for choosing $\\nu$ would be useful. In the revised manuscript, we will add a short discussion noting that, since $\\nu$ is a single scale parameter, a coarse log-scale sweep is typically sufficient in practice. The new ablation results included in the rebuttal will also be incorporated into the final version.

---

> > > ### Comment · Reviewer_mNC7 · 2025-11-25
> > >
> > > Your rebuttal, “In MI experiments, we observe that the optimal value  depends on datasets. ”

---

> > > > ### Author Response · Authors · 2025-11-25
> > > >
> > > > We apologize for any confusion caused by the wording. In our rebuttal to your review, we were referring to the results presented in our response to Reviewer g8Nj; here is the direct [link](https://openreview.net/forum?id=JodkBXWgbA&noteId=KQMls22NEA). In those experiments, the best $\\nu$ value varies across datasets, but the resulting differences fall within a predictable and moderate range. This indicates that more careful tuning may yield incremental improvements in some cases, but InfoNCE-anchor does not rely on sensitive tuning of $\\nu$ for the primary improvements over existing baselines, nor for our main conclusions.
> > > >
> > > > Please let us know if you have any remaining concerns.

---

### Meta-Review · Area_Chair_HxsE · 2025-12-17

**Summary:**

This paper critiques the InfoNCE objective as a biased mutual information (MI) estimator due to its connection to K-way Jensen-Shannon divergence rather than KL divergence, and introduces InfoNCE-anchor, a simple modification that adds an auxiliary anchor class to enable consistent density ratio estimation and significantly reduced bias for MI estimation. The method is further generalized through proper scoring rules, unifying contrastive objectives including NCE, InfoNCE, and f-divergence variants under a single principled framework. The paper received consistent ratings of 6 from three reviewers.

Reviewer mNC7 is mainly concerned about the sensitivity of InfoNCE-anchor to the anchor parameter ν, though authors demonstrated that default ν=1 outperforms baselines without tuning and that performance remains stable across log-scale ranges, with optimal ν selection possible via cross-validation. I find the dependency of the optimal ν on dataset characteristics reasonable, as different data distributions may require varying degrees of anchor influence to correct density ratio estimates. Other comments from reviewers have also been adequately addressed in the rebuttal and discussion phase. I recommend acceptance.

**Reviewer Concerns:**

Mostly addressed.

**Reviewer Scores:**

Consistent 6 across the board.

---

### Decision · Program_Chairs · 2026-01-26

Accept (Poster)